# Multi-omic characterization of allele-specific regulatory variation in hybrid pigs

Jianping Quan [1,2,3,4,9], Ming Yang[5,6,9], Xingwang Wang[1,7,9], Gengyuan Cai[1,3,7,9], Rongrong Ding[1,2,4,6], Zhanwei Zhuang[1,7], Shenping Zhou[1,7], Suxu Tan[2], Donglin Ruan[1,7], Jiajin Wu[1,3,7], Enqin Zheng[1,3,7], Zebin Zhang[1,3,7], Langqing Liu [1,3,7], Fanming Meng[1,8], Jie Wu[1,7], Cineng Xu[1,7], Yibin Qiu[1,7], Shiyuan Wang[1,7], Meng Lin[1,3], Shaoyun Li[1,7], Yong Ye[1,7], Fuchen Zhou[1,7], Danyang Lin[1,7], Xuehua Li[1,7], Shaoxiong Deng[1,7], Yuling Zhang[1,7], Zekai Yao[1,7], Xin Gao[5], Yingshan Yang[1,7], Yiyi Liu[1,7], Yuexin Zhan[1,7], Zhihong Liu[5], Jiaming Zhang[1,7], Fucai Ma[1,7], Jifei Yang[1,7], Qiaoer Chen[1], Jisheng Yang[1], Jian Ye[4,6], Linsong Dong[4,6], Ting Gu[1,7], Sixiu Huang[1,3], Zheng Xu[1,7], Zicong Li[1,7], Jie Yang [1,3,7] ✉, Wen Huang [2] ✉ & Zhenfang Wu[1,4,6] ✉

Hybrid mapping is a powerful approach to efficiently identify and characterize genes regulated through mechanisms in cis. In this study, using reciprocal crosses of the phenotypically divergent Duroc and Lulai pig breeds, we perform a comprehensive multi-omic characterization of regulatory variation across the brain, liver, muscle, and placenta through four developmental stages. We produce one of the largest multi-omic datasets in pigs to date, including 16 whole genome sequenced individuals, as well as 48 whole genome bisulfite sequencing, 168 ATAC-Seq and 168 RNA-Seq samples. We develop a read count-based method to reliably assess allele-specific methylation, chromatin accessibility, and RNA expression. We show that tissue specificity was much stronger than developmental stage specificity in all of DNA methylation, chromatin accessibility, and gene expression. We identify 573 genes showing allele specific expression, including those influenced by parent-of-origin as well as allele genotype effects. We integrate methylation, chromatin accessibility, and gene expression data to show that allele specific expression can be explained in great part by allele specific methylation and/or chromatin accessibility. This study provides a comprehensive characterization of regulatory variation across multiple tissues and developmental stages in pigs.

It is well established that regulatory variation of genes, which changes abundance and spatiotemporal distribution of gene expression as opposed to forms of genes, contributes to phenotypic diversity within and between species[1]. For example, while mutations in coding regions are more likely to cause phenotypes, the vast majority of sequence variants associated complex diseases in humans are non-coding[2]. Many of these noncoding variants are regulatory in nature, affecting different aspects of gene expression including transcription, splicing, transport, and translation, presumably by altering regulatory sequence elements[3,4]. Importantly, expression quantitative trait locus (eQTLs) are enriched for trait-associated variants identified in GWAS (genome-wide association studies)[5], suggesting that these variants contribute to phenotypic variation by regulating gene expression.

---

The genetic architecture of regulatory variation in gene expression is highly dynamic and influenced by cell types and tissues, physiological states, and environments. For example, *cis* eQTLs exert cell-type specific effects by changing sequences of *cis*-regulatory elements[6,7]. In addition, analyses in multiple tissues in a large cohort identified both shared and tissue-specific expression and splice QTLs[8]. Analyses in multiple tissues in a large cohort identified both shared and tissue-specific expression and splice QTLs[8]. Moreover, regulatory genetic architecture is dynamic in blood transcriptomes of aging humans[9] and in primary dendritic cells when exposed to pathogens[10].

There are several ways to characterize the genetic basis of dynamic regulatory variation. Population scale profiling of genome-wide gene expression followed by mapping of eQTLs has proven highly effective but requires large sample sizes[11]. Gene expression profiling in eQTL mapping studies typically ignores the allelic origin of expressed RNAs although allele-biased or specific expression is prevalent when gene expression is regulated by *cis* eQTLs[12]. Allelic imbalance of expression is caused by *cis* sequence divergence between the paternal and maternal alleles and its presence indicates the presence of *cis*-regulatory variation. Therefore, it can be used to efficiently identify divergent gene expression regulated by *cis* variation. For example, by comparing expression originated from the two alleles in an interspecific F1 hybrid of Drosophila, genes subject to *cis* and *trans* regulation were identified, providing a genome-wide characterization of the regulatory landscape[13,14]. A special case of allele-specific expression (ASE) is the parent-of-origin effect, where expression is biased towards a particular allele according to its paternal or maternal origin. Many studies have leveraged ASE in hybrids to identify parent-of-origin effects in mice[15], humans[16], pigs[17,18], and interspecific hybrids (donkey)[19]. However, most studies of regulatory genetic variation focused on steady-state RNA abundance. Few studies have assessed genetic variation at multiple levels in the context of different tissues and developmental stages.

In this study, we carried out one of the most comprehensive studies characterizing allele-specific regulatory variation in any species. We performed eight reciprocal crosses between two genetically and phenotypically divergent pig breeds (Duroc and Lulai), collected replicate samples from fullsibs in four different tissues across four developmental stages, and assayed three layers of genomic variation (methylation, chromatin accessibility, and RNA abundance), amounting to a total of 16 whole genome sequencing of parental genomes, 48 whole genome bisulfite sequencing, 168 ATAC-Seq and 168 RNA-Seq samples. This dataset allowed us to obtain a comprehensive and high-resolution characterization of and important insights into the regulatory landscape of gene expression in pigs.

## Results

### Experimental design

To obtain a comprehensive global view of the *cis*-regulatory landscape in pigs, we crossed pigs from two genetically divergent breeds (Duroc and Lulai) reciprocally, sampled tissues from the F1 hybrids at different developmental stages, and profiled three layers of genomic features including steady-state RNA abundance, chromatin accessibility, and DNA methylation (Fig. 1). Duroc is a major commercial breed known for its superior growth performance, while Lulai is a synthetic breed derived from crosses between Laiwu pigs and Large White pigs[20], which possess desirable meat quality traits, including high intramuscular fat content[21]. We performed four Duroc female × Lulai male crosses and four reciprocal crosses between Lulai females and Duroc males (Fig. 1). In the first Duroc × Lulai cross, three female and three male fetuses at gestational age day 40 were collected, and their brain, liver, muscle, and placenta attached to each fetus were collected. The same tissues from the same fetal stage were also collected in a reciprocal Lulai × Duroc cross (F40, Fig. 1). In the second Duroc × Lulai cross and its reciprocal Lulai x Duroc cross, brain, liver, muscle, and placenta tissues were collected from three female and three male

fetuses at gestational age day 70 (F70, Fig. 1). These two fetal stages (gestational age 40 and 70 days) represent the two major muscle fiber formation waves in fetal development of pigs. In addition to the two fetal stages, we also collected tissues (brain, liver, muscle) for newborns (D1) and market weight adults (D168). In each of these crosses, three female and three male fullsibs were used for tissue sampling, representing three biological replicates within each sex (Fig. 1). In total, we collected 4 (tissues) x 6 (fetuses) x 2 (crosses) x 2 (stages) = 96 tissue samples for the two fetal stages and 3 (tissues) x 6 (animals) x 2 (crosses) x 2 (stages) = 72 for the two postnatal stages for a total of 168 tissue samples.

We performed genomic assays on nucleic acids extracted from these tissues and their parents. DNA sequencing was performed using DNA extracted from the ear tissue of all 16 parents to obtain their whole genome sequences. Strand-specific RNA-Seq was performed for all 168 tissues in the hybrids to obtain genome-wide gene expression data. ATAC-Seq was performed for all 168 tissues to obtain chromatin accessibility data. Finally, whole genome bisulfite sequencing was performed for DNA from all tissues in 70-day fetuses ($n = 48$) to obtain DNA methylation data. This dataset represents one of the most comprehensive sampling to understand the genetic basis of *cis*-regulatory variation in pigs.

### Construction of individualized genomes and transcriptome annotations

The domestic pig reference genome assembly was based on a Duroc pig, which may lead to bias while mapping sequence reads derived from the Duroc and Lulai alleles in the hybrids[22]. To alleviate this problem, we attempted to sequence all 16 parental generation pigs used in this study to an average depth of 16.5X (Supplementary Data 1) but one of the Duroc females failed sequencing in two separate DNA extractions. We called DNA variants using the GATK in the 15 successfully sequenced pigs. On average, we discovered 6.66 M non-reference SNP genotypes (homozygous alternative alleles or heterozygous) in the Duroc pigs but 12.55 M in the Lulai pigs. In addition, an average of 2.42 M and 3.40 M non-reference indel genotypes were discovered in the Duroc and Lulai pigs respectively (Supplementary Data 1). The sharp difference in non-reference genotypes between Duroc and Lulai pigs can be attributed to the genetic divergence between the Duroc and Lulai breeds and the fact that the reference genome was from a Duroc pig. Indeed, the genome-wide Fst value between these two breeds was 0.312, and there was substantial local variation across the chromosomes (Supplementary Fig. 1a). Furthermore, identity by state (IBS) distance between the animals clearly separated the two breeds into two distinct clusters (Supplementary Fig. 1b), which was also confirmed by a principal component analysis (PCA) (Supplementary Fig. 1c).

Many studies use alleles at known SNP positions to call allele-specific coverage for either DNA or RNA and methods are available to mitigate mapping bias[22]. However, this may lead to inconsistent results when there are multiple SNPs in the same genes. We developed an approach based on assignment of reads to the alleles from which they are derived (Fig. 1c). To do so, we first generated individualized genomes for all animals in the parental generation. We replaced reference alleles with homozygous alternative alleles, including both SNPs and indels, and lifted coordinates of transcript annotations to the individualized genomes. The resulting genomes have different lengths and transcript coordinates. We imputed genotypes for the Duroc individual that failed sequencing using monomorphic alleles in the Duroc breed with the assumption that this animal shared the same genotypes with others from the same breed if they all share the same alleles. We indexed the reference genome assembly and the 16 individualized genomes using BWA for subsequent DNA sequence mapping and using HISAT2 in the presence of the individualized transcriptome for RNA sequence mapping.

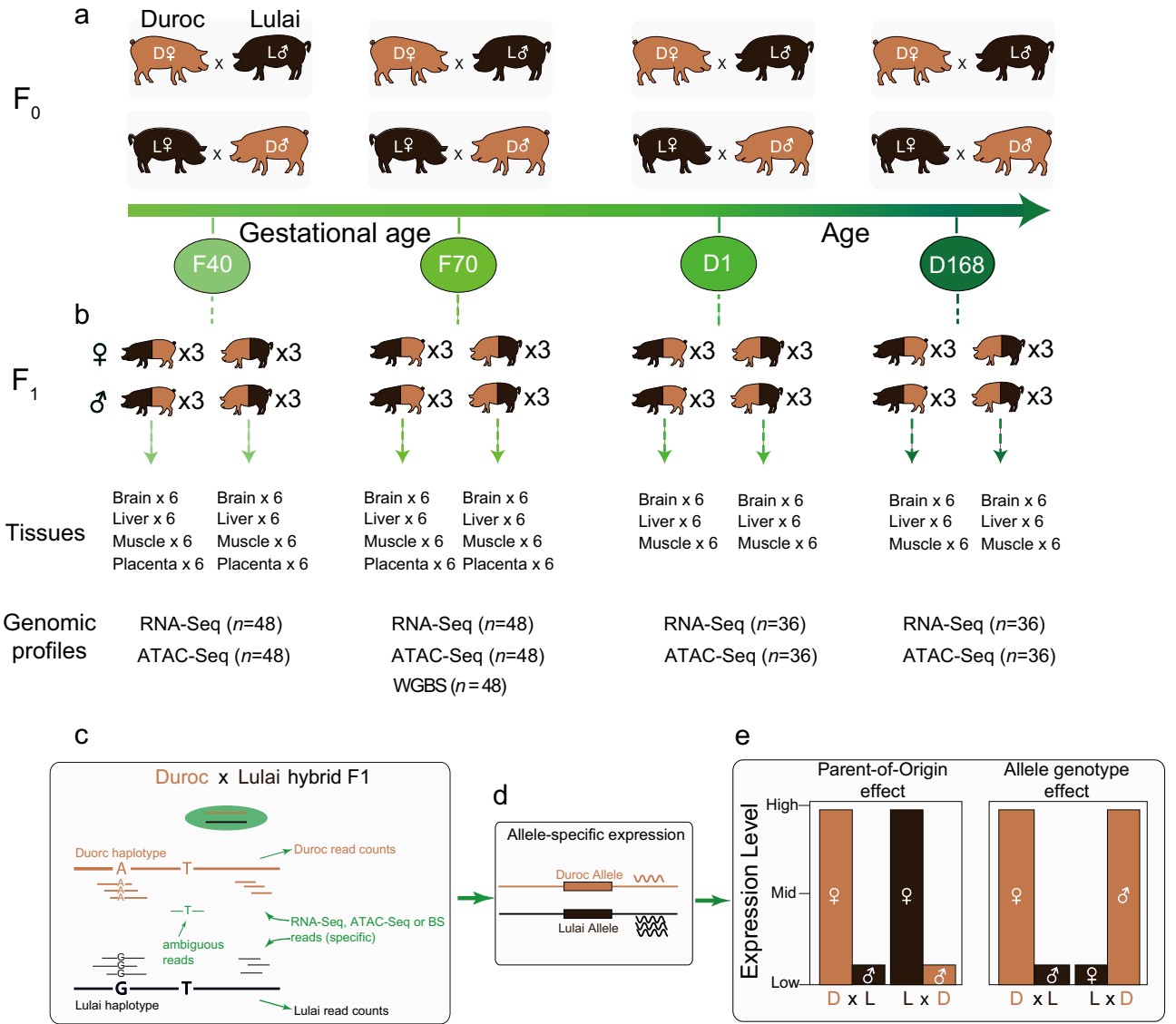

**Fig. 1 | Integrated experimental design and analytical strategies for comprehensive characterization of allelic expression in reciprocal cross pigs. a** Duroc pigs and lulai pigs underwent reciprocal crosses at various developmental stages. **b** Overview of tissue sample size and sequencing contents across developmental stages. **c** Genome assignment of sequencing data in hybrid offspring. The sequencing reads were mapped to individualized genomes, and the origin of reads is determined based on mismatches with informative variants in the personalized genome. If a read matches with an A base, it is attributed to duroc pigs; if it matches with a G base, it is attributed to lulai pigs. If there are no informative variants at the matching position, the origin of the read remains unclear. **d** Quantification of allele expression. **e** Allelic expression comparison and ASE evaluation across tissues and developmental stages. If the expression level of the maternal allele is higher, regardless of whether the maternal parent is duroc or lulai pig, the gene is classified as a POE gene. If the expression level of the duroc allele is higher, regardless of whether duroc is used as the maternal or paternal parent, the gene is classified as an AGE gene.

## Diversity and dynamics of the pig transcriptome

We first analyzed the RNA-Seq data without distinguishing their breed origin in the F1 hybrid animals, including a total of 168 samples (Fig. 1), to an average depth of 52.2 M paired-end fragments (Supplementary Data 2). To identify potential sample misidentification, we called variants from the RNA-Seq data and clustered samples based on their RNA-Seq derived genotypes. Tissues from the same individuals formed tight clusters followed by samples from fullsibs for the same cross (Supplementary Fig. 2). We removed one sample from one F1 individual who did not cluster with other samples from the same individual and another 9 samples from three F1 individuals (all samples from these individuals) who did not cluster with their fullsibs (Supplementary Fig. 2). Genetic distance estimates based on SNP genotypes called from RNA-Seq reads confirmed that these 10 samples were either contaminated or misidentified.

We counted reads mapped to each gene based on the Ensembl annotation using featureCounts in the Subread software suite[23]. PCA using log2 transformed TPM (transcripts per million) value revealed that the brain gene expression program was distinct from other tissues, which explained the majority of variation in the first principal component (Fig. 2a). Other tissues were also clustered when the samples were projected to lower principal components space (Fig. 2b). To identify genes that are dynamic across tissues and developmental stages, we fitted a model including effects of tissue, developmental stage, and their interaction, as well as sex and family effects as covariates using edgeR. We first tested the effect of tissue by developmental stage interaction on gene expression for genes that had counts per million (CPM) >= 3 in at least six samples. Among the 15,716 genes tested, 15,527 had a significant tissue-by-developmental stage interaction effect (FDR = 0.05), suggesting that for almost all genes the

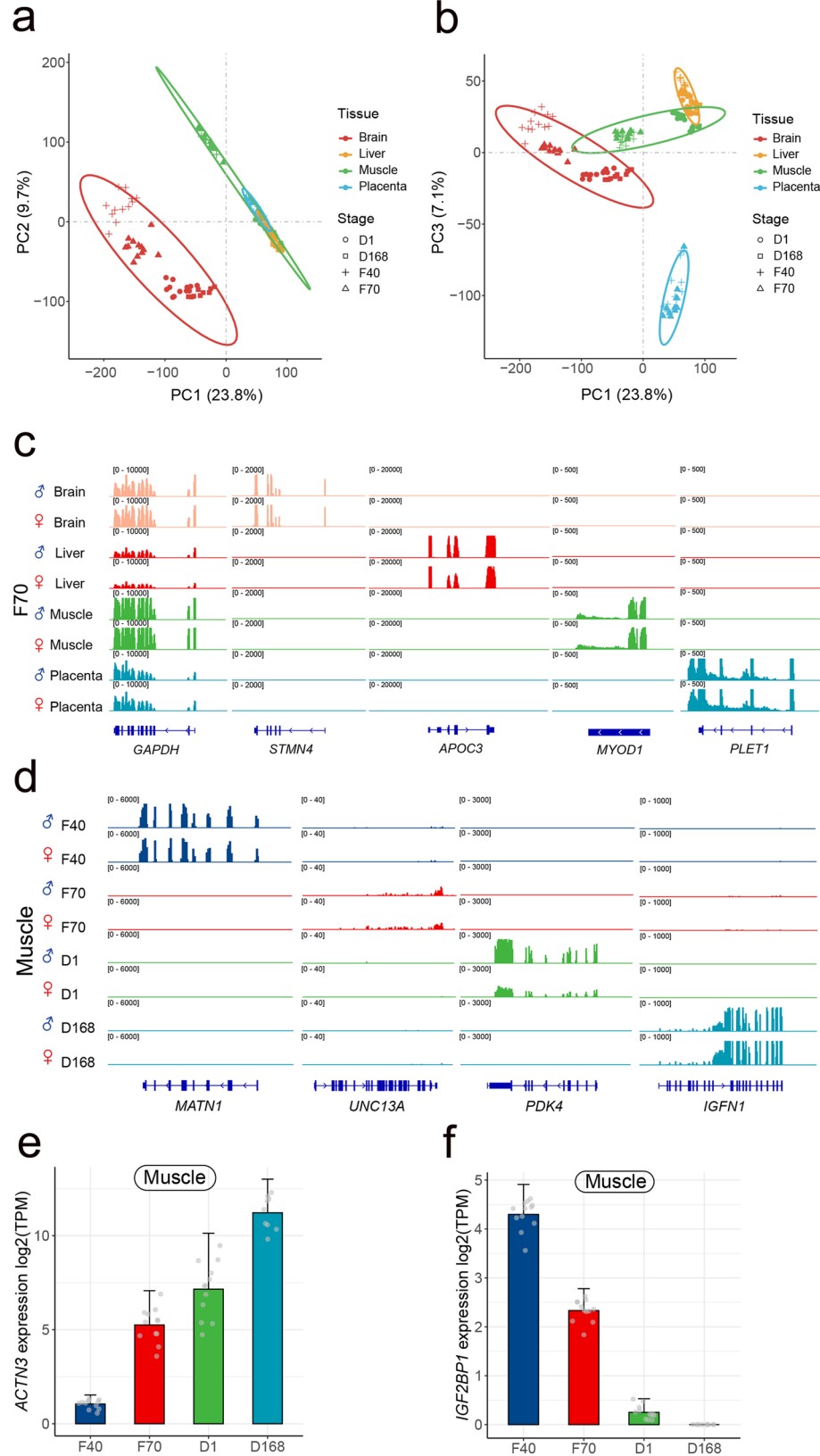

dynamics of gene expression during development are variable across tissues or equivalently, the tissue-specific expression pattern changes during development. Therefore we stratified subsequent analyses according to either tissue or developmental stage.

To identify tissue-specific genes, we fitted a linear model with tissue and sex to the model in each developmental stage separately

using edgeR. We identified tissue-specific genes by requiring that expression of the gene in the reference tissue was at least 16-fold higher than all other tissues. A total of 385, 179, 130, and 268 genes were identified as exhibiting tissue-specific expression in the brain, liver, muscle, and placenta, respectively in all developmental stages (Supplementary Fig. 3). For example, while the housekeeping gene

**Fig. 2 | Tissue and stage specificity of gene expression. a** Principal component analysis (PCA) of gene expression data across various tissues based on the first and the second principal components. The numbers within parentheses represent the variance explained by each principal component. Red, yellow, green, and blue dots represent brain, liver, muscle, and placental tissue samples, respectively. Circular, square, cross, and triangular points represent samples from the D1, D168, F40, and F70 periods, respectively. The ellipses are drawn based on a 95% confidence level for a multivariate t-distribution. **b** PCA of gene expression data based on the first and third principal components. **c** IGV snapshot for RNA-Seq signal in pig tissues at the *GAPDH* locus (a housekeeping gene), as well as several genes with tissue-specific activity. The height of the bars represent the density distribution of RNA-Seq reads around gene. **d** IGV snapshot for RNA-Seq signal of several genes with developmental stage-specific activity in pig muscle. **e** The average expression of *ACTN3* at different developmental stages in muscle (*n* = 48). Each gray dot represents a sample. The up error bars depict the standard deviation of the gene expression. **f** The average expression of *IGF2BP1* at different developmental stages of in muscle (*n* = 48). Each gray dot represents a sample. The up error bars depict the standard deviation of the gene expression. Source data are provided as Source Data file.

*GAPDH* was highly expressed in all tissues, Stathmin 4 (*STMN4*), Apolipoprotein C3 (*APOC3*), Myogenic Differentiation 1 (*MYOD1*), and Placenta Expressed Transcript 1 (*PLET1*) were exclusively expressed in brain, liver, muscle, and placenta respectively (Fig. 2c), consistent with previous reports[24]. Gene ontology enrichment analysis of the tissue specific genes also revealed enrichment of biological processes that are relevant to the respective tissues (Supplementary Data 3).

In addition, in each tissue, we identified stage-specific genes by requiring that expression of the gene in a stage was at least 16-fold higher than all other stages. We identified 59, 97, 89, and 12 genes that exhibited stage-specific expression in the brain, liver, muscle, and placenta, respectively (Supplementary Data 4). For example, the genes *MATN1*, *UNC13A*, *PDK4*, and *IGFN1* were specific to day 40 and 70 fetuses, 1 day and 168 days old pigs, respectively (Fig. 2d). Notably, there were much fewer genes with stage-specific expression than tissue-specific expression, suggesting that the tissue specificity of gene expression was determined early in development and was persistent. This also explained the fact that tissue was the dominant factor in explaining variation in PCA (Figs. 2a, b). We further identified 27 distinct patterns of developmental dynamics of gene expression in each tissue, including monotonic up and downregulation and other more complex patterns (Supplementary Fig. 4). We found monotonic changes to be generally the most frequent patterns. For example, actin 3 (*ACTN3*) showed an obvious monotonic increase in muscle tissue during development, with adult pig (D168) showing by far the highest expression (Fig. 2e). *ACTN3* is only expressed in type-II muscle fibers[25]. The high expression of *ACTN3* in the fast twitch muscle fiber is needed to meet the rapidly developing exercise capacity requirements of piglets, which went from slow and clumsy to normal movement within a few weeks after birth[26]. In addition, the gene Insulin-like growth factor 2 mRNA binding protein 1 (*IGF2BP1*) decreased monotonically during development (Fig. 2f), which appeared to be a conserved pattern in animals, including in poultry[27].

Our results clearly demonstrated the enormous diversity and dynamics of the pig transcriptome across tissues and developmental stages, a result that has been consistently found in complex eukaryotes[28,29].

## Allele-specific expression landscape across tissues and developmental stages

Our experimental design that uses hybrids of genetically divergent pig breeds allows us to efficiently characterize the landscape of allele-specific expression. On average, each pair of the parental animals contained 4,484,545 short allelic variants that were homozygous to alternative alleles in the two parents. Among these genome-wide variants, 220,467 (average on the 7 sequenced pairs) overlapped with exons and were considered informative with the exception of the cross where the female parent's sequence was imputed. These exonic variants enabled us to assess the allele-specific expression of on average 14,960 genes by assigning sequenced reads to their respective parental or breed origins.

Instead of inspecting allelic coverage at known SNP sites, we developed a pipeline where we mapped RNA-Seq reads to both individualized genomes of the parents separately and assigned reads to each genome by matching divergent alleles in these reads. This procedure produced read counts for each gene for the paternal and maternal haplotypes separately, allowing us to harness existing read count-based RNA-Seq analysis engines. We first assessed overall coverage for each of the parental haplotypes. Among the 152 samples that were retained, the vast majority of samples had nearly equal paternal and maternal reads (Supplementary Fig. 5). We filtered out samples that had paternal reads constituting more than 55% or less than 45% of total reads, all of which were placenta tissues when maternal contamination was an issue. Furthermore, we compared the proportion of reads mapped to the maternal allele for each SNP, and the proportion of reads mapped to the maternal genome for the gene where the SNP resides. There is a strong bias towards unequal allele representation by SNP-based allele counts than by gene-based read counts. Other than agreement along the diagonal, the slope of the points is substantially smaller than 1, suggesting that SNP-based analysis tends to significantly overestimate allele bias (Supplementary Fig. 6). These results strongly indicate that our read-based approach is sufficient to overcome the bias introduced by SNP-based analysis.

Within each tissue and developmental stage, we fitted a model including the parent-of-origin effect (POE), allele genotype effect (AGE), maternal genome (MG) effect, and sex, the latter two of which were covariates which we do not formally perform statistical inference. The POE was defined as the effect by the maternal or paternal origin of the allele, and the AGE was defined as the effect by the breed origin of the allele (Fig. 1e). Among the 14 tissue–developmental stage combination, we were able to examine 5872–9588 genes for effects of POE and AGE and identified 8−27 genes with significant paternally biased POE (FDR = 0.05, Fig. 3a, Supplementary Data 5), 5−21 genes with significant maternally biased POE, and 20−54 genes with significant Duroc biased AGE and 20−38 genes with significant Lulai biased AGE (FDR = 0.05, Fig. 3a, Supplementary Data 6), amounting to a total of 179 unique POE genes and 394 unique AGE genes. Among the four tissues, the most where POE (*n* = 101) and AGE (*n* = 213) genes were identified was the brain.

The POE genes clearly showed a tendency to cluster on chromosomes, suggesting that they may be under the control of the same *cis*-regulatory element (Fig. 3b). We compared our POE genes with previously identified regulatory elements in pigs and found substantial overlaps (Supplementary Table 1). Relative to randomly sampled genomic regions, the proportion of pair-wise distances smaller than 1 Mb between POE genes was 2.1 times higher (Fig. 3b). The most extreme POE genes are imprinted genes, which often are tissue or developmental-stage specific[30]. Among the 43 known imprinted genes according to the gene imprint database (https://www.geneimprint.com/), 37 could be assessed for their imprinting status in any of the tissue and developmental stage. We found 11 of these 37 genes had statistically significant POE and 9 other that had strong but not statistically significant POE (expression bias of the paternal allele > 75% or <25%) (Fig. 3c). The discrepancy between the imprinting status in this study and others may be due to imprinting being limited to specific tissues and developmental stages. Nevertheless, the genes with strong POE effects constituted the majority (20/37) of the assessed known imprinted genes in the present study. In addition to these known imprinted genes, we identified 168 genes with statistically significant POE effects, among which 17 genes had a 95%/5% average expression

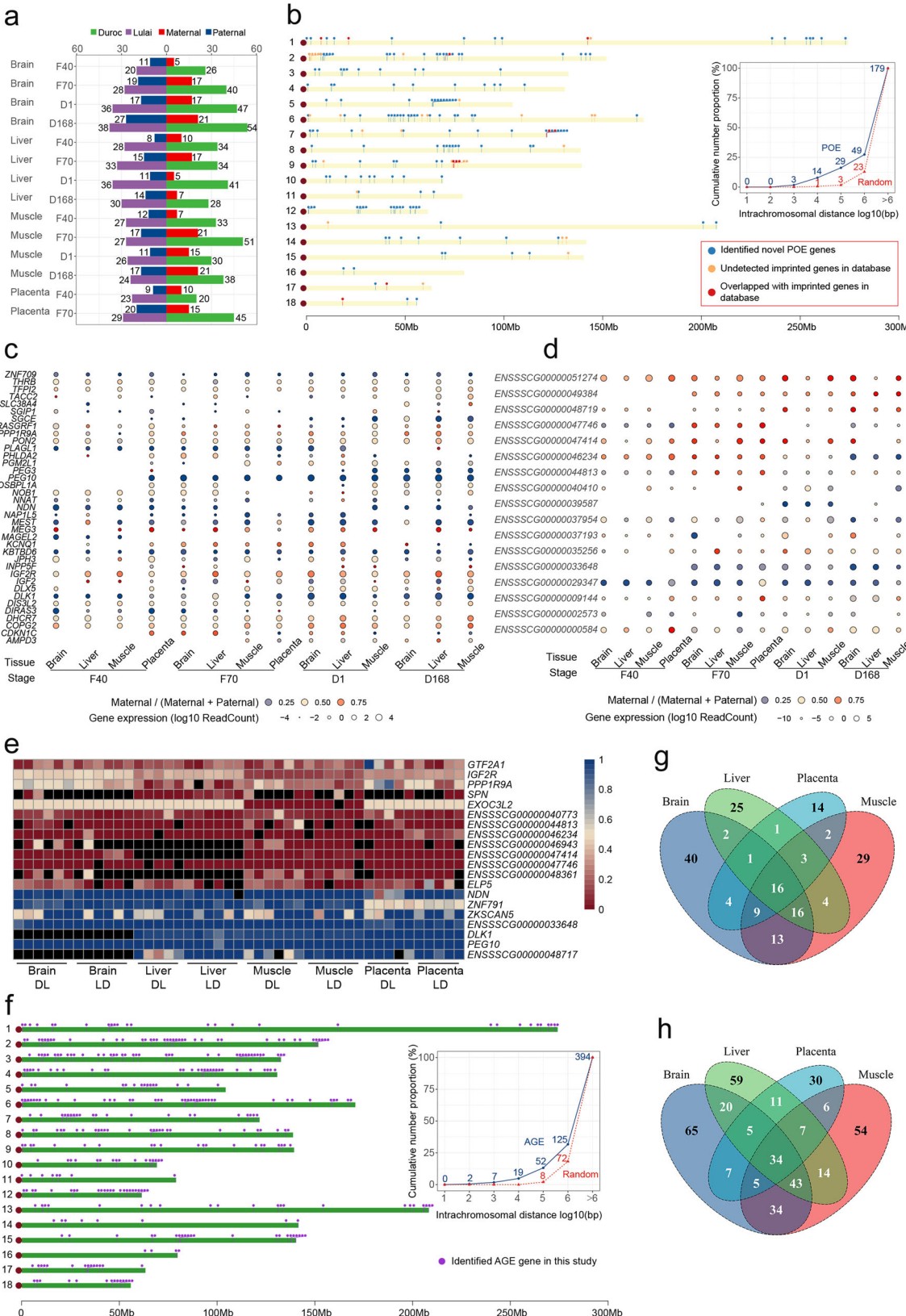

bias in the same tissue in at least one developmental stage (Fig. 3d). We considered these are imprinted genes (Supplementary Data 7). To demonstrate the complex pattern of genomic imprinting in pigs, we showed here several examples. First, *IGF2R* is a known imprinted gene whose imprinting is tissue depenent[17,31]. Consistent with this knowledge, we showed that *IGF2R* was biallelically expressed in brain but

maternally expressed in liver, muscle, and placenta in all stages. Furthermore, we found an imprinted gene *EXOC3L2*, which was expressed in all tissues but only paternally imprinted in muscle (Fig. 3e). Another known imprinted gene is *PEG10*, which was broadly imprinted in all tissues in this study (Fig. 3e). Furthermore, many lncRNAs showed imprinted expression pattern limited to specific tissue

**Fig. 3 | Allele specificity of gene expression. a** The number of genes exhibiting parental-of-origin effects (POE) and allele genotype effects (AGE) was determined for each tissue-stage context group. **b** POE genes distribution in pig genome. Each dot represents a gene, and short vertical lines indicate the location of the gene on the chromosome. The line chart on the right shows the distance in a chromosome selected at random, the number of conventional genes and POE genes within the random distance. **c** The heatmap shows the expression and expressed direction of imprinted genes discovered by previous works in this study. Blue color in circle indicates paternal allele bias and red color in circle indicates maternal allele expression bias. The size of the circles indicated the amount of gene expression, which was transformed according to log10 reads count. **d** The heatmap shows the expression and expressed direction of imprinted genes that were identified in this study. **e** The heatmap shows tissue specificity of several POE genes. Red (0)

indicates that maternal origin allele was expressed and paternal allele was imprinted, and blue (1) indicates that maternal origin allele was imprinted and paternal origin allele was imprinted. **f** AGE genes distribution in pig genome. Each dot represents a gene, and short vertical lines indicate the location of the gene on the chromosome. The line chart on the right shows the distance in a chromosome selected at random, the number of conventional genes and AGE genes within random distance. **g** The POE genes that were shared among four tissues. Numbers above venn diagram of each tissue indicate the count of POE genes that are either specifically identified in one tissue or overlapping across multiple tissues. **h** The AGE genes that were shared among four tissues. Numbers above venn diagram of each tissue indicate the count of AGE genes that are either specifically identified in one tissue or overlapping across multiple tissues. Source data are provided as Source Data file.

and developmental stage, including *ENSSSCG00000048717*, *ENSSSCG00000046943*, *ENSSSCG00000047414* and *ENSSSCG0 0000048361* (Fig. 3e). Additionally, our investigation revealed 14 genes exhibiting inconsistent parental allelic expression biases that switched direction across developmental stages (Supplementary Data 5). To understand biological functions that are most affected by the POE genes, we performed pathway enrichment analysis and found that paternally biased POE genes were enriched for many amino acid and fatty acid metabolism pathways, while maternally biased POE genes were enriched for basal transcription factors, bile secretation and PPAR signaling pathways (Supplementary Data 8).

Unlike POE genes, AGE genes were more widely distributed across the entire genome (Fig. 3f). The proportion of pairwise distance smaller than 1 Mb between AGE genes was 1.7 times higher than randomly selected genomic regions (Fig. 3f). This suggested that their regulatory mechanisms are more localized, which is expected as the hybrid mapping design will only identify *cis*-regulatory AGE effects. We found 16, 29, 26 out of the 179 POE genes were shared among four, three, and two tissues, respectively, but 34, 64 92 out of the 394 AGE genes shared among four, three, and two tissues, respectively (Fig. 3g, h). Tissue specificity was more apparent among POE than AGE genes. Pathway enrichment analysis revealed that the Duroc-biased AGE genes were highly enriched for metabolic pathways, including protein digestion and absorption, butanoate metabolism, fatty acid degradation, and metabolism, while Lulai-biased AGE genes were enriched for genes related to immune responses (Supplementary Data 9). These may be related to the highly desirable metabolic traits of the Duroc breed and disease resistance and resilence traits of the Lulai breed.

Taken together, the allele specific RNA abundance analysis in the reciprocal crosses with multiple tissues and developmental stages identified a substantial number of genes whose expression are under the control of either a parent-of-origin effect or allele genotype effect. These effects are often tissue and developmental stage-specific and may be related to the phenotypic divergence of the two pig breeds.

## Dynamics of chromatin accessibility across tissues and developmental stages

To understand the epigenetic basis of the dynamics of gene expression across tissues and developmental stages, we performed ATAC-Seq of nuclear DNA extracted from the same tissue samples where RNA expression was profiled (Fig. 1). We obtained data from all samples except for 5 that failed library preparation QC (quality control), including one liver and four placenta samples.

First, we mapped ATAC-Seq reads to the reference genome and called peaks using MACS2 in samples combined within each tissue, developmental stage, sex, and cross, with the assumption that peaks from biological replicates are largely common and high sequencing depth is beneficial for peak calling. Peaks were then merged to obtain a catalog of ATAC peaks across all conditions. Peak heights were quantified for each sample with respect to the common catalog of peaks. In

general, there was a strong enrichment of open chromatin around TSS (transcription start site) (Fig. 4a), testifying to the ability of ATAC-Seq to identify peaks that are expected to be accessible. We performed a PCA and hierarchical clustering on the matrix of peak heights for all samples and observed clear clusters of tissues within each developmental stage (Figs. 4b,c, Supplementary Fig. 7).

We identified 346,793 peaks across all conditions, with an average size of 468 bp (200 to 5060 bp). The combined length of these peaks accounted for 6.5% of the pig genome. The number and length of peaks for each condition showed significant variations. We annotated the chromatin-accessible regions for genomic features and found that 14% of the peaks were located in promoters (<3k upstream of TSS) while 34% were located in distal intergenic and 45% in intronic regions (Fig. 4d). Given the small proportion of promoter regions of the whole genome, this represented a remarkable enrichment of open chromatin near TSS. When considered in each tissue and stage independently, 40 day fetal livers had the highest proportion of peaks overlapping with promoters (40%). Consistent with the distinct clusters in PCA as well as gene expression, the tissues and stages showed a substantial diversity in terms of open chromatin in genomic features (Supplementary Fig. 8).

To demonstrate the concordance between chromatin accessibility and gene expression, we focused on the tissue specificity of these two layers of variation. We identified tissue-specific open chromatin regions in each tissue and asked whether they overlap with genes expressed in the same tissues. Remarkably, in all four tissues, the overlap between tissue-specific open chromatin and expressed genes was highly significant (Table 1). Furthermore, the patterns of tissue-specific open chromatin and gene expression largely agreed (Fig. 4e). These results suggested that tissue specific chromatin accessibility is a key driver in dictating tissue specific gene expression programs. For example, while there were clear ATAC peaks in all tissues for the housekeeping *GAPDH* gene, open chromatin was restricted to the tissues where the genes were expressed for non-housekeeping genes (Figs. 2c, 4f). Additionally, in genes exhibiting stage-specific expression during development, the concordance between open chromatin and gene expression patterns was also strong (Supplementary Fig. 9). These observations further underscore the significance of developmental stage-specific chromatin accessibility in regulating stage-specific gene expression programs.

## Allele-specific chromatin accessibility is associated with allele-specific gene expression

The enrichment of overlap between tissue-specific open chromatin and genes with tissue-specific expression patterns suggested that open chromatin at least partially drives gene expression. To quantitatively assess whether allele-specific chromatin accessibility causes allele-specific gene expression, we estimated allelic bias in ATAC-Seq peak intensities and compared them with allelic bias in gene expression. We summed all reads assigned to the paternal or maternal allele associated with each gene over multiple peaks and used the same procedure as in

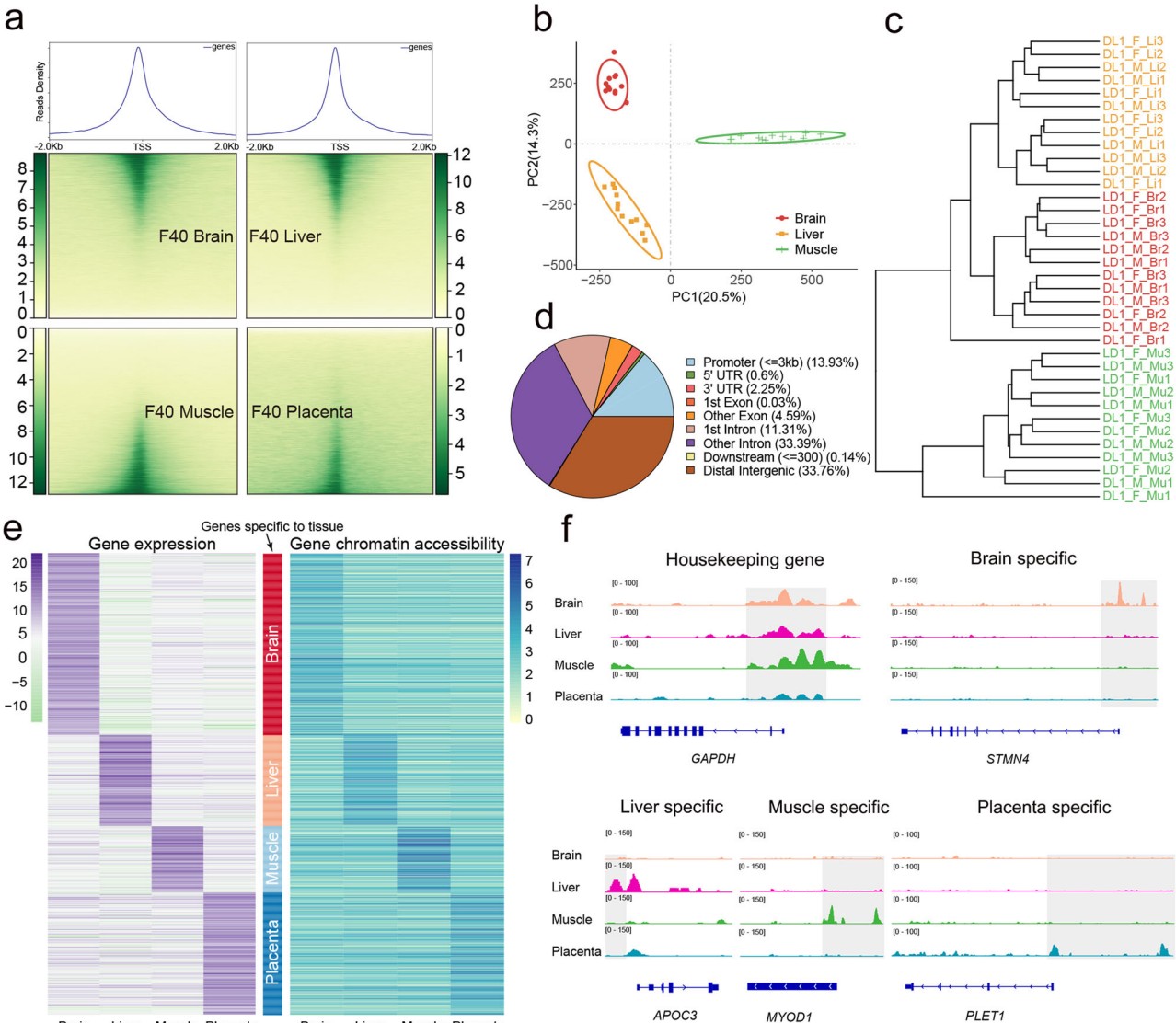

Fig. 4 | Tissue specificity of chromatin accessibility. a Heatmaps depicting normalized ATAC-seq signal at all TSS, sorted by signal intensity. Signal shown for brain, liver, muscle and placenta. b PCA of normalized ATAC-seq signal in consensus open chromatin identified in brain, liver, and muscle tissue at D1 stage. c Cluster tree of samples based on normalized ATAC-seq signal in consensus open chromatin. Each leaf node represents a sample. d Distribution of pig consensus open chromatin relative to genomic features. Because peaks often span multiple features, peaks were categorized based on 1 bp overlap with features in the following order: first as promoter (3 kb upstream of TSS), 5' UTR, 3' UTR, exon,

intronic, and if no features were overlapped, peaks were considered intergenic. e The heatmap of tissue-specific expression gene shows the gene expression and chromatin accessibility of gene promoter. The different colored bars in the middle of the two heatmaps represent genes exhibiting tissue-specific expression in various tissue. f ATAC-Seq signal in pig tissues at the housekeeping gene and several genes with tissue-specific activity. The height of the peaks represent the density distribution of ATAC-Seq reads around gene. The shade denotes the approximate area harboring the identified different reads distribution in chromatin accessibility in this study. Source data are provided as Source Data file.

**Table 1 | The consistent analysis between tissue-specific expression genes identified in RNA-Seq and ATAC-Seq**

|  | Tissue-specific peaks annotated gene number (fold change >= 2) | Tissues-specific expression gene number (fold change >=16) | Overlapped genes number | Enrichment score | *P* value |
|---|---|---|---|---|---|
| Brain | 1245 | 388 | 149 | 8.1 | 0.00001 |
| Liver | 109 | 180 | 15 | 15.4 | 0.00001 |
| Muscle | 311 | 131 | 35 | 17.2 | 0.00001 |
| Placenta | 1849 | 268 | 100 | 4 | 0.00001 |

Note: *P*-value was calculated through permutation based test.

gene expression to identify allelic bias that was due to either POE or AGE. In total we identified 1,670 and 1,990 genes (fold change >= 2, FDR = 0.05) whose chromatin accessibility has significant POE and AGE effects, respectively (Supplementary Fig. 10).

We asked whether allele-specific gene expression was associated with allele-specific chromatin accessibility. Across all four tissues and all stages, 25 out of 179 genes (*P* = 0.0004) with expression POE also had chromatin accessibility POE, and 42 out of 394 genes (*P* = 0.03) with

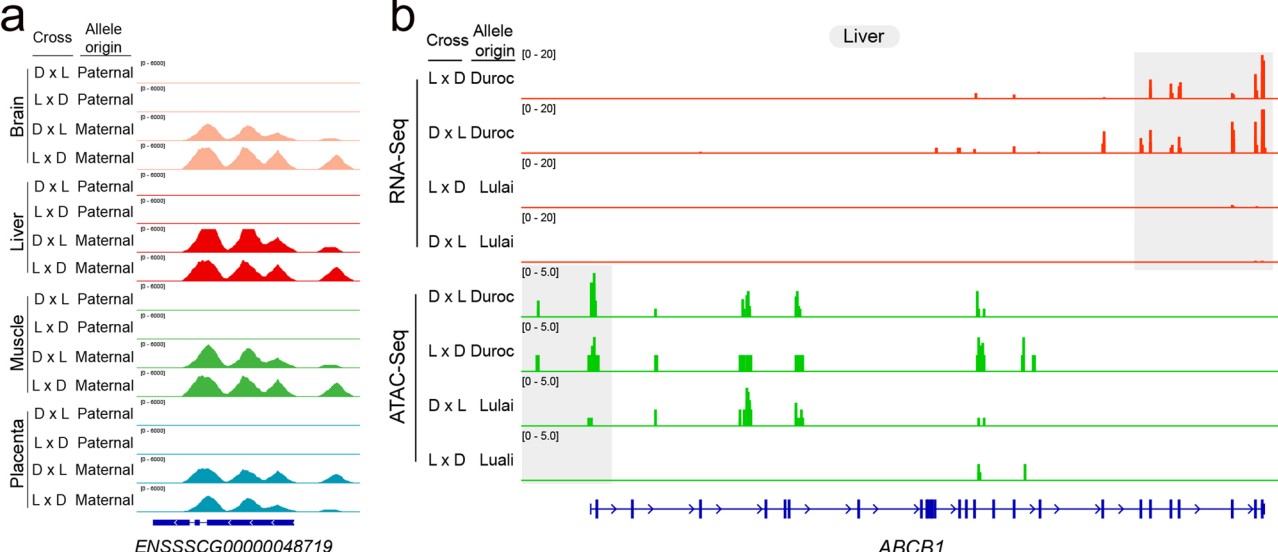

**Fig. 5 | Allele specificity of gene expression is associated with allele specificity of chromatin accessibility. a** ATAC-Seq signal of POE gene *ENSSSCG00000048719* with different parental orgin from the reciprocal cross between Duroc and Lulai pigs. The height of the peaks represent the density distribution of RNA-Seq and ATAC-Seq reads from different parental origin around gene. The D × L and L × D stand for hybrid Duroc (female) × Lulai (male) and Lulai (female) × Duroc (male), respectively. **b** ATAC-Seq signal of AGE gene *ABCB1* with different allele genotype from the reciprocal cross between Duroc and Lulai pigs. The height of the peaks represent the density distribution of RNA-Seq and ATAC-Seq reads from different breeds around gene. The shade denotes the approximate area harboring the identified different reads distribution in gene expression and chromatin accessibility in this study.

expression AGE also showed AGE in their chromatin accessibility, both of which were significantly enriched (Supplementary Fig. 11). When bias scores in gene expression and chromatin accessibility were correlated, in the 14 tissue/stage combinations, all but two for POE (ranging between −0.01 and 0.54) and all but one for AGE (ranign between −0.01 and 0.47) genes had a positive correlation between allelic bias in chromatin accessibility and gene expression (Supplementary Fig. 12). These results suggest that differential chromatin accessibility between the two alleles has the tendency to but does not fully drive differential gene expression in the same direction. In some cases, however, allele specific expression and chromatin accessibility showed a remarkable degree of agreement. For example, a POE gene *ENSSSCG00000048719* that was paternally imprinted had accessible chromatin only on the maternal allele (Fig. 3d, Fig. 5a). In addition, *ABCB1* was a Duroc biased gene with an AGE effect and its chromatin near the TSS of this gene also exhibited Duroc biased accessibility (Fig. 5b).

**Dynamics of DNA methylation across tissues**

We next considered genome-wide DNA methylation, which precedes transcription and may also influence chromatin accessibility. We focused on the F70 fetuses and performed whole genome bisulfite sequencing for all tissues at this stage, totaling 48 samples. First, we mapped bisulfite sequencing reads to the reference genome and extracted sequenced bases on the CpG context to evaluate cytosine methylation in each sample using the Bismark software. We computed methylation level at each site as the proportion of cytosines that were not converted by bisulfite treatment. PCA and hierarchical clustering were performed on the matrix of methylation levels across all 48 samples and 36,090,067 CpG sites with coverage >= 10 in at least 8 samples of each tissue (Figs. 6a, b). The analyses revealed distinct clusters by tissues. In the four tested tissues, the brain and placenta tissues had the highest and the lowest global methylation, respectively (Fig. 6c). While in brain and muscle CpGs were generally highly methylated, methylation in liver and placenta was much more variable across the genome (Fig. 6d).

We then asked if methylation preferentially occurred in certain genomic features. We found methylation within non-promoter regions

to be relatively stable with the highest average methylation found in introns and 3' UTRs (untranslated regions). Methylation in promoters was generally low and showed a clear valley in proximity to transcription start sites (Fig. 6e). Importantly, we found significant negative correlation between methylation in promoters and cognate gene expression (Fig. 6f), suggesting that methylation in promoters serve as a general negative regulator of gene expression. Likewise, there was a negative association between the magnitude of chromatin accessibility and the extent of methylation in all tissues. Except for placental, the correlation between methylation and chromatin accessibility was stronger than that between methylation and gene expression (Fig. 6g), consistent with the hypothesis that the effects of methylation on gene expression may be mediated by chromatin accessibility. In genes exhibiting tissue-specific expression, both chromatin accessibility and methylation near them showed a similar pattern of groupings according to tissues (Supplementary Fig. 13). For example, the *STMN4* gene specifically expressed in brain exhibited prominent ATAC peaks, which were accompanied by noticeably lower methylation levels in the promoter region compared to other tissues (Fig. 6h).

**Effects of allele-specific DNA methylation on allele-specific expression and chromatin accessibility**

To further understand the mechanisms underlying allele-specific gene expression, we investigated the impact of allele-specific methylation on allele-specific expression and chromatin accessibility. We aligned bisulfite sequencing reads to the parental genomes and assigned reads to their parental origins based on variant alleles they contained. This enabled us to assess allele-specific cytosine methylation in the CpG context. Within each F70 tissue separately, we excluded CpG sites with coverage fewer than 5 reads in more than 2 samples out of the six from either the paternal or maternal alleles in each direction of cross. A total of 436,381 to 709,328 CpGs were retained for subsequent analysis across the four tissues. These CpGs were assigned to 7828, 8239, 7557, and 7334 genes, as well as to the promoter regions of 5284, 5693, 5052, and 4762 genes in the brain, liver, muscle, and placenta, respectively.

To investigate whether the POE genes had allele-specific DNA methylation according to parental origin, we estimated methylation

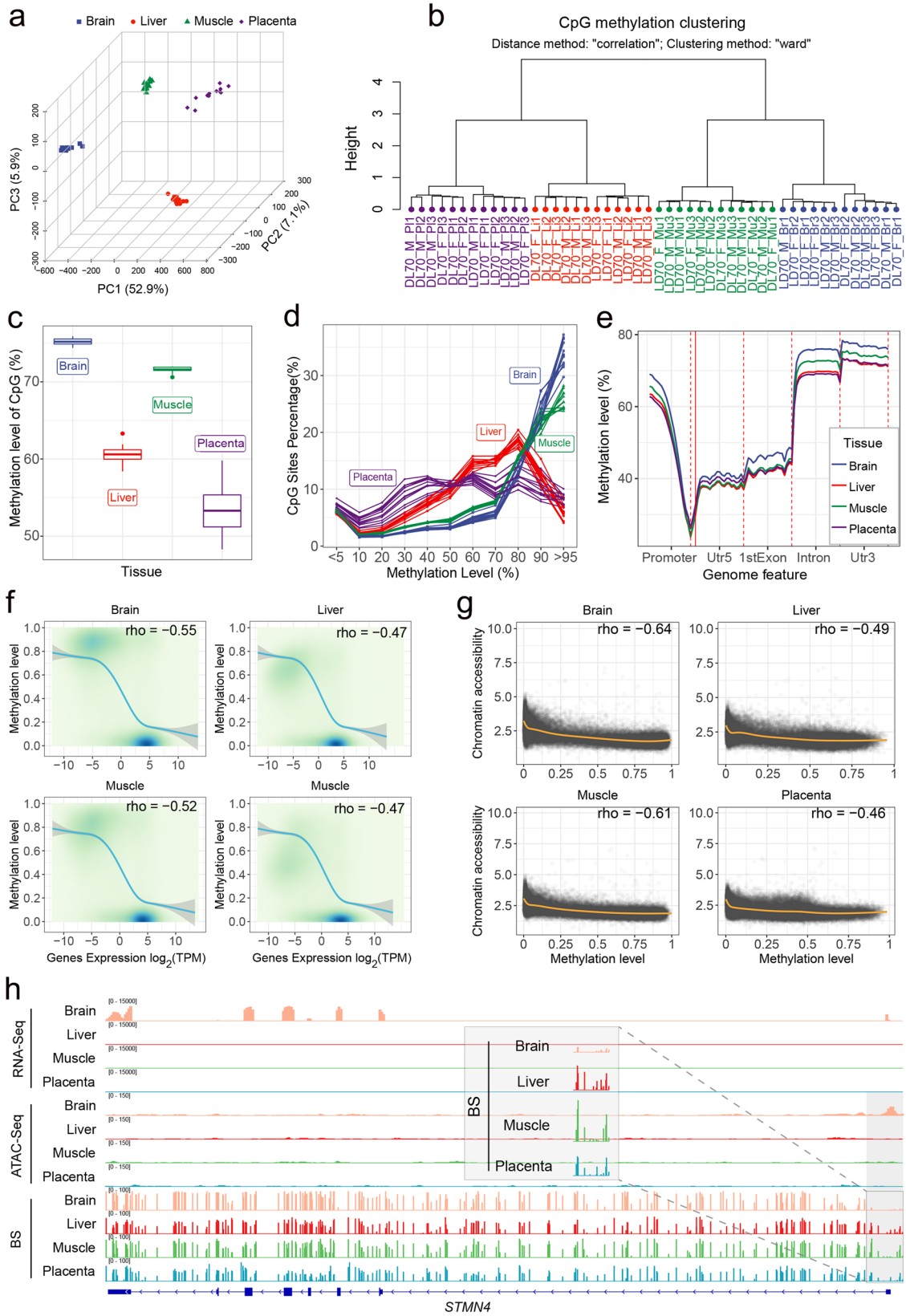

proportion in CpG sites found in the promoter regions of 8, 7, 10, and 11 POE genes in the brain, liver, muscle, and placenta, respectively (Supplementary Data 10). We asked whether CpGs in promoters of these genes were methylated on the allele where gene expression was suppressed. On a genome-wide level, overall methylation in promoters of genes was highly similar between the maternal and paternal

chromosomes (Supplementary Table 2). The bias between parental alleles in methylation in the promoters for 5 of the 13 genes was in agreement with the assumption that higher methylation leads to lower gene expression (Supplementary Data 10). For example, the maternally imprinted gene *PEG10* had a high methylation score near the 5′ end of the gene on the maternal allele but low methylation score on the

**Fig. 6 | DNA methylation across pig tissues and its association with gene expression and chromatin accessibility. a** Three-dimensional PCA plot depicting the methylation levels of CpG sites in each tissue, where each point signifies a distinct sample. **b** CpG methylation clustering with the distance method "correlation" and clustering method "ward" from the clusterSamples function of the R package Methylkit. The meaning of sample label is as follows: DL and LD stand for hybrid Duroc (female) × Lulai (male) and Lulai (female) × Duroc (male), respectively. The 70 in the label indicates that the sample is from developmental stage F70. The letters F and M indicate the sex of the sample individual, F for sow and M for boar. Br, Li, Mu and Pl stand for brain, liver, muscle and placenta, respectively. The numbers 1,2,3 refer to the number of biological replicates individual. **c** Methylation levels of CpG sites in four tissues ($n = 12$). The Y axis represents the percentage of methylation levels. Boxplots are represented by minima, 25% quantile, median, 75% quantile, and maxima with data points. **d** The percentage of CpGs

in each methylation level rank in four tissues. **e** The average methylation level for CpG are shown along the promoter, 5' UTR, exon, intron, 3' UTR at four tissues. **f** The correlation between gene expression and mean methylation level of CpGs in gene promoter. The color gradient indicates the density of data points, with darker colors representing higher densities. The smoothing curve represents the relationship between gene expression and their methylation level of promoter, with the shaded gray area around the curve indicating the 95% confidence interval. **g** The correlation between chromatin accessible intensity and mean methylation level of CpGs in peaks. **h** IGV snapshot for transcriptome, chromatin accessibility, and methylation signal arround the brain-specific expression gene *STMN4*. The shade denotes the approximate area harboring the identified different reads distribution in chromatin accessibility and methylation in this study. Source data are provided as Source Data file.

paternal allele in the brain (Fig. 7a). Allelic bias in chromatin accessibility and gene expression were also observed. The paternal allele was much more accessible and expressed at a much high level than the maternal allele, a result consistent with the hypothesis that methylation of the maternal allele of *PEG10* inhibits gene expression through maintaining a closed state of chromatin (Fig. 7a). This pattern has also been observed in another randomly selected imprinted gene *SGCE* (Supplementary Fig. 14).

Furthermore, we investigated whether the AGE genes exhibited differential allele-specific methylation with respect to their Duroc and Lulai origins. After QC, we identified CpG sites within the promoter regions of 33, 32, 42, and 38 AGE genes in the brain, liver, muscle, and placenta, respectively (Supplementary Data 11). Among these, methylation bias in the promoters of 35 out of the 89 genes had increased methylation and reduced gene expression. For instance, the gene *PM20D1* displayed a high methylation score near the TSS of the Duroc allele but a low methylation score on the Lulai allele in the liver (Fig. 7b). Additionally, we observed allelic bias in chromatin accessibility and gene expression, with the Lulai allele exhibiting significantly higher accessibility and expression levels compared to the Duroc allele (Fig. 7b).

Finally, we used a generalized linear model framework to identify individual CpGs that exhibited differential DNA methylation between alleles, including effects of POE and AGE. A total of 22,340, 26,665, 20,158, and 18,386 CpG sites with POE effect were identified in the brain, liver, muscle, and placenta tissues (FDR = 0.05). A total of 128,882, 140,919, 122,082 and 99,718 CpG sites with AGE effect were identified in the brain, liver, muscle, and placenta tissues (FDR = 0.05). A POE or AGE methylation region (poeMR/ageMR) is defined as a genomic interval containing more than 2 consecutive CpG sites showing significant POE or AGE, respectively (adjusted $P < 0.05$). Using a sliding window of 1.5 kb, we identified a total of 3638, 4673, 3238, and 2933 poeMRs in the brain, liver, muscle, and placenta tissues, respectively. These poeMRs were located in 1108, 1382, 982, and 996 genes in the brain, liver, muscle, and placenta tissues, respectively. For example, a poeMR were found in the *IGF2R* gene in the liver, muscle, and placenta tissues, which was consistent with the tissue-dependent parental bias of *IGF2R* in gene expression (Fig. 7c, Supplementary Fig. 15). Furthermore, a total of 26,176, 28,065, 24,758 and 20,553 ageMRs were identified in the brain, liver, muscle, and placenta, which were located in 26, 31, 33 and 34 AGE genes in the brain, liver, muscle, and placenta, respectively.

## Discussion

Allele-specific expression mapping is a powerful approach to map genes under *cis*-regulatory influence by homogenizing cellular environments for the two alleles in hybrid individuals[32,33]. However, there remain many limitations and challenges. First, because parental genotypes are typically unavailable, mapping to a single reference genome leads to bias[34]. We overcame this limitation in this study by sequencing

the parents. Second, allele-specific expression is typically estimated on a per SNP basis. It is challenging to integrate estimates across different SNPs. We developed an approach to assign reads to parental alleles, which enabled us to leverage existing methodologies based on read counts. Third, very few if any studies have integrated multiple layers of genomic variation to characterize the regulatory mechanisms. The present study represents a multi-omic characterization of allele-specific regulatory variation. To this end, we used a reciprocal hybrid design of two divergent pig breeds and comprehensively sampled multiple tissues and developmental stages. This study represents a pioneering effort in comprehensively assessing DNA methylation, chromatin accessibility, and gene expression across diverse tissues and developmental phases within the context of reciprocal hybrids. This approach facilitated a quantitative estimation of epigenomic and transcriptomic signals in an allele-specific manner, culminating in a high-resolution, genome-wide atlas of *cis*-regulatory variation in pigs across three distinct layers.

Our identification of genes exhibiting distinct expression patterns across tissues and developmental stages not only highlights the dynamic nature of gene regulation but also provides valuable insights into the nuanced control of gene expression during development. Consistent with findings in mice by Wang et al.[35], we observed a significantly higher number of tissue-specific genes compared to developmental stage-specific genes, suggesting that tissue specificity of gene expression is established early in development and persists. Moreover, the expression of genes in the pig genome displayed variable patterns of dynamic changes across developmental stages, rather than a simple monotonic change.

To unravel the intricate link between chromatin structure and transcriptional regulation, we employed ATAC-Seq to explore chromatin accessibility throughout the entire pig genome. Approximately 6.5% of the pig genome exhibited accessibility in at least one tissue or developmental stage, consistent with similar findings by Halstead et al.[36]. Our study, similar to Liu et al. in mice[37], demonstrated a noticeable enrichment between open chromatin promoter regions and tissue-specific expressed genes. This strongly suggests that tissue-specific chromatin accessibility plays a pivotal role in dictating tissue-specific gene expression programs. Given the established role of DNA methylation in controlling chromatin organization and accessibility[38,39], we integrated DNA methylation data to elucidate its relationship with gene expression and chromatin accessibility, especially in tissue-specific expression genes. The methylation level in promoters was identified as a general negative regulator of gene expression, with a consistent negative association observed between the magnitude of chromatin accessibility regions and the extent of methylation across all tissues.

When analyzing allele-specific gene expression, we focused on 14,960 genes in the pig genome that contained informative variation between the two breeds, identifying a total of 179 and 394 genes with significant parental (POE) and sequence (AGE) effects, respectively. In

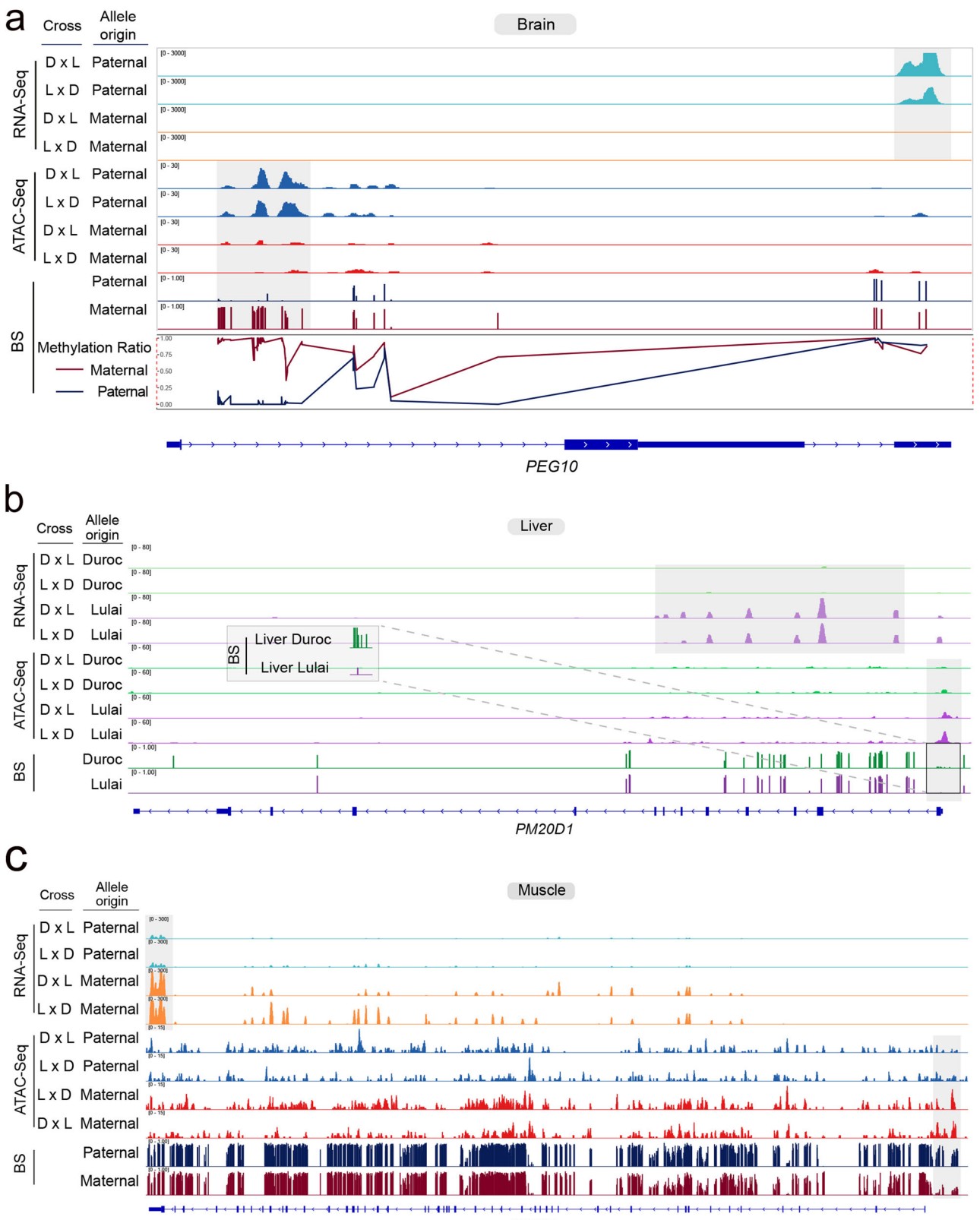

**Fig. 7 | Multi-omic integration of regulatory variation. a** The allelic levels of RNA, chromatin accessibility, CpG methylation, and methylation scores for the imprinted gene *PEG10* are depicted across various hybrid combinations. The methylation ratio at each CpG site is calculated as the number of methylated reads divided by the total reads (methylated and unmethylated) covering that CpG site. The red line signifies the methylation levels of maternal alleles, while the blue line represents the methylation levels of paternal alleles. The x-axis indicates CpG sites, and the y-axis represents the corresponding methylation ratios. The shade denotes the approximate area harboring the identified different reads distribution in gene expression, chromatin accessibility, and methylation in this study. **b** Allelic levels of RNA, chromatin accessibility, and CpG methylation are illustrated for the AGE gene *PM20D1*. A detailed view focuses on the methylation signal within the *PM20D1* promoter region, highlighting allelic distinctions from Duroc and Lulai pigs. **c** Allelic levels of RNA, chromatin accessibility, and CpG methylation are shown for the tissue-dependent imprinting gene IGF2R.

previous studies, using a SNP-based test (as opposed to our read count-based approach) Wu et al.[17] identified 16 imprinted genes in the reciprocal crosses of Duroc and Diannan small-ear pigs, while nine genes showed allele-specific expression in the reciprocal crosses of Korean native pigs and Landrace pigs[40]. Notably, eight putative imprinted genes identified by Wu et al.[17] were also imprinted in our study; three of them were not assessed because they did not contain informative and an five of them were not expressed in sufficiently high level to be reliably tested for POE effect. The high rate of confirmation and a much larger number of POE genes proved that our approach was highly powerful and accurate. Among the identified POE and AGE genes, those with paternal bias in POE were enriched in amino acid and fatty acid metabolism pathways. Conversely, maternally biased POE genes were associated with basal transcription factors, bile secretion, and PPAR signaling pathways. Duroc-biased AGE genes displayed enrichment in metabolic pathways such as protein digestion and absorption, butanoate metabolism, and fatty acid degradation. In contrast, Lulai-biased AGE genes were enriched for genes associated with immune responses. The distinct enrichment patterns of POE and/ or AGE genes when considering their direction of allele biased expression may contribute to the distinct quantitative traits of the two breeds. This underscores the importance of considering the paternal and maternal breeds in selecting pig hybrid combinations. It's notable that a small subset of POE and AGE genes showed a change in direction of their effects across developmental stages. This could be attributed to alteration in the regulation of expression for both POE and AGE genes during development and when context changes, which is not uncommon[41]. Moreover, tissues at different developmental stages may have different compositions of cell types, potentially leading to variation in the direction of POE and AGE in bulk tissues.

To the best of our knowledge, this study provides a comprehensive multi-omic characterization of multiple tissues and developmental stages in pigs. This provides an opportunity to integrate multiple layers of variation to elucidate the cascade of regulatory events. We were able to identify agreement between the allele specific RNA abundance, chromatin accessibility and CpG methylation, offering strong evidence that changes at the epigenetic level can contribute to gene expression variation in a population. Furthermore, we found that the correlation between methylation and chromatin accessibility was stronger than that between methylation and gene expression. This may indicate that methylation has a more direct impact on chromatin accessibility, through which its impact on gene expression is mediated. Nevertheless, larger sample size and deeper sequencing are required for more comprehensive quantitative characterization, especially when involving methylation. While the design in this study can discover genes under *cis* regulation and potentially reveal the regulatory cascade, it offers no information on the regulatory mechanisms with respect to DNA variants. Further high resolution mapping using population scale datasets, high throughput reporter assays, and gene editing are required to causally delineate the *cis* regulatory mechanisms.

In conclusion, this study offers a comprehensive characterization of allele-specific regulatory variation in pigs, encompassing DNA methylation, chromatin accessibility, and gene expression data. We provide a global overview of tissue and developmental specificity in gene expression, allele-specific effects, and the agreement between chromatin accessibility and gene expression. Our integrative approach provides a foundation for further research into the intricate mechanisms underlying gene regulation in pigs, with implications for breeding, genetics, and functional genomics studies.

## Methods

### Ethical statement
All experimental procedures conducted in this study adhered to the ethical standards set by the Institution Animal Care and Use Committee (ACUC) at South China Agricultural University (SCAU) (approval number SCAU#2014-0136).

### Biological sample collection
Two swine breeds, namely Duroc and Lulai, were crossbred to provide heterozygosity for assessing allele specific regulatory variation. The hybrid crosses between these two pig breeds were performed reciprocally, including Duroc (male) × Lulai (female) and Duroc (female) × Lulai (male) (Fig. 1). Offspring samples from the four crosses were collected at four different time periods: 40 day fetuses (F40), 70-day fetuses(F70), 1 day after birth (D1), and 168 days after birth (D168). Brain, liver, muscle, and placenta (only under F40 and F70 periods) were collected from all offspring, and ear tissue samples were collected from all parents. The sexes of the fetuses were determined by detecting the presence or absence of the *SRY* gene, which is exclusively found on the Y chromosome[42]. Tissue samples from three males and three females were collected. For each reciprocal cross, four tissues were collected from F40 and F70 individuals for three females and three males, and three tissues were collected from D1 and D168 individuals for three females and three males. In total, 168 samples were selected for RNA-seq and ATAC-seq. Additionally, 48 samples (tissues from F70 individuals only) were chosen for WGBS (whole-genome bisulfite sequencing), and 16 parent ear samples were used for WGS (whole-genome sequencing) (Fig. 1).

### Nucleic acid extraction, quality evaluation and sequencing
**a. Whole-genome sequencing.** The DNA of each sample was extracted using standard phenol/chloroform nucleic acid extraction protocol outlined in Sambrook and Russell (2006)[43]. Ear tissue samples from 16 parental individuals were used for this purpose. The DNA extracted from these samples was used for subsequent library construction using the TruSeq Nano DNA LT Library Preparation Kit (Illumina Inc., USA) whose purity and size were validated using an Agilent Bioanalyzer 2100 (Agilent Technologies, USA). Subsequently, the libraries were sequenced in paired-end, 150-bp mode using the Illumina HiSeq X10 platform by Novogene (China). One sample was excluded from subsequent analysis due to multiple failed sequencing library construction attempts.

**b. RNA sequencing.** Total RNA was isolated from 168 samples, including brain, muscle, liver, and placenta tissues, using the Trizol protocol. The extraction followed the manufacturer's instructions with minor adjustments. The RNA samples were used for strand-specific RNA library construction and sequencing. To enrich for mRNA, ribosomal RNA was removed from 3 μg of total RNA per sample using the Epicentre Ribo-zero™ rRNA Removal Kit (pig; Epicentre, USA). Strand-specific libraries were generated from the rRNA-depleted RNA using the NEBNext® Ultra™ Directional RNA Library Prep Kit for Illumina® (NEB, USA), following the manufacturer's recommendations. The library products were purified using the AMPure XP system, and their quality was assessed using the Agilent Bioanalyzer 2100 system. The final libraries were sequenced on an Illumina NovaSeq 6000 platform, generating 150 bp paired-end reads.

**c. ATAC sequencing.** To perform ATAC sequencing, the same 168 samples used for RNA extraction were subjected to additional processing. The tissue samples were washed with a 0.09% NaCl solution, followed by grinding into powders using liquid nitrogen. Subsequently, lysis buffer was added to the powders, and the mixture was incubated for 10 minutes on a rotation mixer at 4°C. The cell suspension was then filtered using a 40 um cell strainer and washed with cold PBS (phosphate buffered saline) buffer three times. After inspecting the purity and intactness of the nuclei under microscopy, approximately 50,000 nuclei were allocated for tagmentation using the method described by Corces et al.[44].

The ATAC-seq protocols and Tn5 transposed DNA were purified using AMPure DNA magnetic beads[44]. A qPCR reaction was performed on a subset of the DNA to determine the optimal number of PCR cycles (average of 11 cycles), and the amplified libraries were analyzed on an Agilent Tapestation 2200 (Agilent Technologies) using a D5000 DNA ScreenTape to assess quality by visualizing nucleosomal laddering. Biological replicates were performed in duplicate for all ATAC experiments. The final library was sequenced on the Illumina HiSeq X10 platform with 150 bp paired-end mode.

**d. Whole-genome bisulfite sequencing.** A total of 48 tissue samples collected for the F70 animals were used, and 100 ng of genomic DNA was extracted from each sample according to the procedure described above. To improve library complexity, 0.5 ng of lambda DNA was spiked into each sample. The genomic DNA was then fragmented to sizes ranging from 200 to 300 bp using the Covaris S220 system. Following the manufacturer's instructions, the fragmented DNA was subjected to bisulfite treatment using the EZ DNA Methylation-GoldTM Kit (Zymo Research) to convert unmethylated cytosines to uracils. The bisulfite-converted DNA fragments underwent adapter ligation and indexing PCR using the Accel-NGS Methyl-Seq DNA Library Kit (Swift Accel) to generate indexed libraries. The quality of the resulting libraries was assessed using the Agilent Bioanalyzer 2100 system. Finally, paired-end sequencing was performed on the Illumina platform (Illumina, CA, USA).

### Data preprocessing

**a. Detection of parental variants and individualized genome reference construction.** The raw data obtained from WGS was processed using the fastp tool[45]. This processing step involved removing reads containing adapters, reads with poly-N sequences, and low-quality reads from the raw data, resulting in clean data in fastq format. Subsequently, the clean reads were aligned to the original reference genome (*Sscrofa11.1*) using the BWA MEM software (v 0.7.16a)[46]. To identify and mark duplicate reads, the MarkDuplicates tool in the Picard toolkit (version 2.13.2) was employed. The average sequencing depth and coverage for each sample were calculated using the depth module in Samtools[47] and the genomeCoverageBed module in Bedtools[48] based on the mapped reads. Next, the base quality of the mapped reads was recalibrated using GATK (version 4.1.1.0). For each parental genome, variants were called using the GATK HaplotypeCaller and GenotypeGVCFs tools[49]. A population VCF (variant call format) file was generated by combining GVCFs from all parental samples. The variants present in the autosome and X chromosome were then filtered using the pipeline established by Ding et al. to get the final variants list[50]. The final variants from the VCF file were used to compute the IBS distance between individuals using the PLINK software[51]. This analysis helps to assess the genetic relatedness between individuals based on shared genetic variants. PCA was performed on the final list of variants using GCTA 1.93.2 for all individuals[52]. Furthermore, the genome-wide Fst value between the parental Duroc and Lulai populations was computed using Vcftools software with 100 K window steps[53].

We changed in the reference genome any bases where homozygous non-reference genotypes were observed in an individual, including SNPs and indels but excluding SNPs that overlapped with indels. The incorporation of indels led to coordinate shifts therefore gene annotation in GTF (gene transfer format) was lifted over to the individualized genomes by offsetting cumulative coordinate. This process produced an individualized reference genome and corresponding GTF annotations while retaining all sequence and annotated feature identifications for each sequenced parent.

**b. RNA-seq read alignment and gene expression quantification.** The fastp software[45] was utilized to process raw RNA-seq reads, ensuring

data quality by removing low-quality reads and those containing poly-N sequences. The resulting clean reads were aligned to the original reference genome for pigs using HISAT2[54]. To identify potential sample misidentification, variant calling was performed on the RNA-Seq data, and samples were clustered based on their genotypes derived from RNA-Seq. Samples from the same individuals that did not cluster together were identified and removed from the cohort to ensure sample integrity. Furthermore, to quantify gene expression levels, the featureCounts module in the Subread software was utilized. This module assigned the aligned reads to specific genomic features, such as genes, and quantified the number of reads mapped to each feature. By calculating the read counts assigned to each gene, gene expression levels could be determined. These expression levels were then transformed into TPM values, which account for differences in gene length and library size. The resulting read count matrix was used for subsequent tissue- and stage-specific analysis, while the TPM value matrix was employed for PCA analysis to assess global patterns of gene expression.

To estimate allele specific expression and test for parent-of-origin and allele genotype effects, we mapped clean RNA-Seq reads from each hybrid animal to each of the two individualized parental genomes. Each read was then assigned to one of the two parental genomes by considering its mismatches against each genome. To assign a read to a genome, we required that the read contained more alleles matching the genome than its mate and that at least 80% of the reads overlapped with constitutive exons. We required that the two ends of sequenced fragments did not disagree in read assignments. In other words, one of the two paired-end reads may not be assignable due to lack of informative alleles. After read assignment, we counted number of reads originated from each gene using BEDtools, resulting in a read count table for each of the two parental genomes.

**c. ATAC-seq read alignment, peak calling and quality accessment.** Prior to mapping, low quality bases and residual adapter sequences were trimmed from raw sequencing data using fastp (version 0.21.0)[45]. Trimmed reads were aligned to original reference genome of pig assembly using BWA MEM (version 0.7.17) with default settings[46]. Duplicate alignments were removed with Picard-Tools (version 2.18.2), and mitochondrial and low quality alignments (q < 30) alignments were removed using Samtools (version 1.6)[47]. The unique BAM files were used to check the insert size distribution of sequenced fragments that could be used to evaluate ATAC-seq data quality. The insert size distribution is expected to have a periodicity of approximately 200 bp, which corresponds to the size of fragments protected by one nucleosome (Supplementary Fig. 16). Furthermore, the deduplicated bam files from biological replicates samples within the same cross, developmental stage and sex were merged, and were used for narrow peaks calling stringently by MACS2[55] with options "--nomodel --shift −100 --extsize 200 -q 0.01". All narrow peaks were further merged by Bedtools (version 2.30.0)[48] to obtain a preliminary comprehensive set of "consensus" peaks that accounted for accessible chromatin regions (ACRs) in any tissues and developmental stages. The bedgraph files, which produced from peak calling process, were further used to calculate peak score after deducting noise by the MACS2 bdgcmp module, and converted to bigwig file that was visualized by IGV through bedGraphToBigWig (version 2.8). Peak abundances were computed by the featureCounts module in the subread software based on ATAC-seq reads and ACRs coordinate. Peak abundance matrix based on the original reference genome was used for tissue- and stage- specific ACRs analysis. Genome-wide accessible chromoatin signal was normalized by RPKM in 50 bp windows using the bamCoverage function from the deepTools suit. The computeMatrix and plotHeatmap functions from the deepTools suit was further used to visulaize the signal at TSS loci with options "computeMatrix reference-point −beforeRegionStartLength 2000

–afterRegionStartLength 2000 –skipZeros" and "plotHeatmap --colorMap YlGn --whatToShow 'heatmap and colorbar'".

The ATAC-seq reads from each individual were mapped both individualized genomes of the parents. After removing duplicate alignments, mitochondrial reads, and low-quality reads, the reads that overlapped with informative SNPs or INDELs were selected and assigned to one of the two parental genomes. Reads assigned to each parental genome were counted for peak abundance using Bedtools[48], resulting in a paternal and a maternal abundance for each peak in each individual.

**d. WGBS reads alignment and methylation level statistics.** In the analysis of genome-wide bisulfite sequencing reads, the following steps were performed: 1) Trimming of raw reads: The raw reads of each sample were trimmed using cutadapt (version 2.10) software[56]. This step involved removing adapter sequences, low-quality bases (q < 20), and short reads (L < 50). 2) Bisulfite-converted reference genome generation: The original reference genome was transformed into a bisulfite-converted version, with C-to-T and G-to-A conversion. The converted genome was then indexed using bowtie2[57]. 3) Read mapping: The clean reads were mapped to the original reference genome of the pig using the bismark software (version 0.22.3)[58] with the following parameters: --score_min L,0,−0.6, -X 700 –dovetail. 4) Removing duplicate reads: Duplicate reads were removed using the deduplicate_bismark module in the bismark software. Only uniquely mapped reads were used for the summary of sequencing depth and coverage of methylcytosine. 5) Sodium bisulfite non-conversion rate: The sodium bisulfite non-conversion rate was calculated as the percentage of cytosine sequenced at cytosine reference positions in the lambda genome. 6) Extraction of CpG methylation: The unique mapping reads were used to extract methylation information for all CG contexts using the *bismark_methylation_extractor* module. The analysis focused only on CpG methylation. 7) CpG methylation matrix: The CpG methylation information, including coverage numbers of methylated and unmethylated reads, was extracted from the methylation extractor result according to the final CpG list. These data were combined to generate a CpG methylation information matrix for all samples. 8) CpG coverage number filtering: CpG coverage numbers of samples in each tissue were filtered, requiring a coverage number greater than 10 in at least 10 out of 12 samples. SNPs identified in the WGS pipeline described above were removed from the union CpG list to generate the final CpG list for subsequent analysis.

The trimmed reads from each individual were also aligned to their respective parental modified bisulfite-converted genomes using the Bismark software. Duplicate reads were removed using the *deduplicate_bismark* module in Bismark. The clean BAM file generated was intersected with variants to extract informative reads. These reads were then sorted by name and assigned to either paternal or maternal genome origin. Subsequently, the methylation status of each cytosine site was calculated based on parental genomes, and the coordinates of each cytosine site were converted back to the original genome coordinates such that the methylation status from both parental genomes can be compared. Bases within inserted sequences in either genome, not present in the reference genome, were excluded. The counts of methylated and unmethylated reads for each cytosine site in each individual were tallied. To align with the previous analysis on the original genome, emphasis was placed on the CpG context of cytosine in subsequent analyses, leading to the removal of CHG and CHH contexts. Ultimately, by aggregating the methylation sites from multiple samples, a list of CpG methylation levels for all individuals was generated based on their paternal and maternal genomes. The coverage number under the parental genome of each CpG was employed to filter out CpGs with low coverage, retaining only those with a coverage number greater than 5 in at least 4 samples within each cross.

**Statistical analysis**

**a. RNA-seq data analysis.** The gene expression matrix, generated based on read counts, which included samples from different tissues and stages, underwent a filtering step. Genes were filtered to retain those with Counts per Million (CPM) values greater than 3 in at least 6 samples. The model was fitted as follow: ~ tissue + stage + tissue:stage using edgeR[59]. The effect of the tissue by developmental stage interaction on gene expression was tested to identify genes that exhibited dynamic expression patterns across tissues and developmental stages. The analysis revealed that the dynamics of gene expression during development varied across tissues, indicating changes in tissue-specific expression patterns during development. To identify tissue-specific genes, a linear model incorporating tissue and sex as factors was fitted separately for each developmental stage. Tissue-specific genes were identified by comparing the expression of a gene in the reference tissue with that in other tissues, requiring a minimum 16-fold higher expression in the reference tissue[60] and a false discovery rate (FDR) value less than 0.05.

The gene expression matrix was generated from each individual's RNA-seq reads aligned to their respective paternal and maternal modified genomes using the pipeline described above. To assess the mapping bias between the paternal and maternal genomes, the total read counts derived from each parental genome were compared for each sample. A normal fluctuation range of 0.45 to 0.55 in the read count ratio (maternal genome original reads count / paternal genome original reads count) was considered acceptable. Samples with a parental ratio exceeding this range were excluded from subsequent statistical analysis. The R package edgeR was utilized to identify ASE genes, which exhibit parent-of-origin-dependent effects (POE) or allele-genotype-dependent effects (AGE), the model was fitted as follow: ~POE + AGE + MG + Sex. The MG and SEX represent the maternal effect and sex effect, respectively. In this study, genes showing expression effects due to parental origin were referred to as POE genes, while genes influenced by allele genotype were referred to as AGE genes. Both POE and AGE genes were included in the definition of ASE genes. The identification of ASE genes was performed on subset samples grouped by tissue at each stage. Initially, genes were filtered based on a CPM value greater than 1 in at least 6 samples, and the significance cutoff was set at an FDR of less than 0.05. Additionally, alleles showing imbalanced expression were required to differ by more than 4 times between different origins (maternal vs. paternal or Duroc vs. Lulai), as measured by the adjusted expression of each gene in the parental genome. To uncover potential ASE genes in various contexts (tissue combined with stage), a gene was aslo considered ASE if it met the significant cutoff condition (FDR < 0.05) and exhibited at least a 1-fold change in expression in two contexts. The Kyoto Encyclopedia of Genes and Genomes (KEGG) pathway analysis and gene ontology (GO) enrichment analysis were conducted using KOBAS 3.0 tool[61], with the list of genes involved in the ASE analysis serving as the background.

**b. ATAC-seq data analysis.** According to the read count matrix generated above, PCA of signal in consensus peaks was performed using the R *prcomp* function to get an overview of different samples. The genomic distribution of ACRs and associated genes was confirmed by ChIPseeker[62]. According to the annotation result that peaks were assigned to the corresponding genes, peaks located at distal intergenic of gene were filtering out. Whereafter, the read count in peaks that belong to a gene was sum together as the chromatin opening intensity of the gene. The glmLRT function in the edgeR package was used to identify genes with different ACRs between tissues and between developmental stages. The significant cutoff was set as |logFC| > 2 and FDR < 0.05. The tissue-specific or stage-specific chromatin accessible genes detected in this ways were used to compare with tissue-specific or stage-specific expression genes that identified by RNA-seq data to get omics correlation between RNA-seq and ATAC-seq. The bigwig file

of ATAC-seq data was visualized on the Integrative Genomics Viewer (IGV) to investigate the chromatin accessibility changes.

To further explore the chromatin accsessible status around ASE gene, the clean reads of offsprints were aligned to modified parental genome using BWA MEM. Duplicate alignments, mitochondrial and low-quality alignments were removed first. Then, the reads that could be clearly distinguished parental origin were separated into maternal and paternal files, which were further used to count the peak intensity according to peak list coordinate. Finally, we obtained a matrix of peaks with reads count from maternal and paternal genomes of each sample. And according to the genome annotation results of peaks, we integrated the peak-reads count matrix into the gene promoter region-reads count matrix. The model was fitted as follow: -POE + AGE + MG + Sex using edgeR to analyze POE and AGE of chromatin accessibility regions. In addition, we also focused on parental origin chromatin accessibility in the promoter region of POE/AGE gene to explore the potential mechanism of POE/AGE phenomenon. The maternal/paternal (M/P) ratio was calculated of each POE or AGE gene according to gene expression and according to chromatin accessible intensity of gene number. When the M/P ratio obtained from gene expression and chromatin accessible intensity of gene promoter showed the same bias direction, these POE/AGE genes would be considered to be affected by POE and AGE chromatin accessibility in the gene promoter region. Several representative POE/AGE genes will be shown in IGV software to visualize this association between gene expression and chromatin accessibility of gene regulation.

**c. WGBS data analysis.** The PCA and correlation analysis based on the methylation level of each CpG site in all samples were performed using the methylKit package[63] in R. The cluster tree was generated using clusterSamples function with the 'ward' clustering method in methylKit package. The methylation level of each CpGs was defined as the coverage number of methylated reads (C) divided by the total coverage number of methylated reads (C) and unmethylated reads (T) at the same positions. The average methylation level of a sample was calculated using the total coverage number of methylated reads divided the total coverage number of reads that covered CpGs. The proportion distribution of CpGs at each methylation level bin (low methylation <5%, high methylation >95%, and 5%-95% with every 10%) was calculated using CpGs number at different methylation level divided by total CpGs number that read coverage number greater than 10. The CpGs were annotated to genome features, including 5' UTR, 3' UTR, promoter intron, 1st Exon and gene body, using R package clusterProfiler[64]. The average methylation level of each genome feature or chromatin accessible region (peak) was computed as methylated reads divided by the total reads number of all covered CpGs in each genome feature or peak. Association between gene expression and methylation level of genome feature, including promoter and gene body, were evaluated by Spearman correlation coefficient. In addition, the association between chromatin accessibility of peaks and methylation level in that peak were also evaluated by the Spearman correlation coefficient. In gene level, the relationship between gene expression, gene chromatin accessibility (removed peaks in distal intergenic) and methylation level of gene promoter were visualized by the pheatmap package in R after zscore transforming of methylation level.

Furthermore, the coverage of filtered CpGs across all individuals was employed to calculate the methylation level for each allele. Subsequently, the allele-specific methylation analysis was conducted using the EdgeR package for detecting differential methylation at individual CpG sites. The generalized linear model (glm(cbind(unm, m) - sex + po + ag + mg, family = "binomial")) was utilized to evaluate methylation differences at each CpG site and generate p-values for parental-of-origin effect, allele genotype effect, maternal genotype effect, and sex effect. Subsequently, FDR correction was applied to adjust the p-values for each CpG. A parental-of-origin effect methylation region (poeMR) or allele genotype effect methylation region (ageMR) was defined as a genomic interval with consecutive CpG sites exhibiting consistent POE or AGE CpG events. The poeMRs or ageMRs were delineated as regions containing more than 2 consecutive CpG sites displaying significant POE or AGE (adjusted $P < 0.05$) within a sliding window of 1.5 kb.

## Reporting summary

Further information on research design is available in the Nature Portfolio Reporting Summary linked to this article.

## Data availability

All raw high-throughput sequencing data generated in this study were submitted to the GSA database under accession numbers CRA014032, CRA013965, CRA013959 and CRA013956. Source data are provided with this paper.

## Code availability

All the computational scripts and codes used in this study are available on GitHub repository: https://github.com/JianpingQuan/ASEanalysis and https://doi.org/10.5281/zenodo.11204941.

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

## Acknowledgements

This study was financially supported by a Natural Science Foundation of China project (31972540 to J.Y.), a key Technologies R&D Program of Guangdong Province project (2022B0202090002 to Z.F.W.), a Local Innovative and Research Teams Project of Guangdong Province (2019BT02N630 to J.Y.), a USDA-NIFA HATCH project (MICL02560 to W.H.), a USDA-NIFA AFRI project (2021-67021-34149 to W.H.), and Michigan State University AgBioResearch (to W.H.). We also would like to express our gratitude to the WENS Foodstuff Group Co., Ltd. for their

valuable assistance in the breeding of experimental animals. Their support and collaboration were essential in conducting this study.

## Author contributions

Z.F.W., Jie Y., and W.H. conceived and designed the study presented in this paper. Z.F.W., M.Y., Jian Y., G.Y.C. and L.S.D. contributed to the management of experimental pigs. J.P.Q., X.W.W., R.R.D., Z.W.Z., S.P.Z., J.W., C.N.X., Y.B.Q., D.L.R. and S.Y.W. contributed to sample collection. F.C.Z., D.Y.L., X.H.L., S.X.D., Y.L.Z., Z.K.Y., X.G., Y.S.Y., Y.Y.L., Y.X.Z., Z.H.L., J.M.Z., F.C.M., Jifei Y., Q.E.C., Jisheng Y. contributed to sample nucleic acid extraction. D.L.R., J.J.W., Y.Y., S.Y.L. and M.L. managed and maintained the data and performed the quality control. E.Q.Z., F.M.M., L.Q.L., Z.B.Z., T.G., S.X.H., Z.X., and Z.C.L. contributed server maintenance for data analysis. J.P.Q., W.H. and S.X.T. conducted the analyses. J.P.Q. and W.H. wrote the paper. Z.F.W., Jie Y., W.H. and L.Q.L. revised the manuscript. All authors discussed results, read and approved the final paper.

## Competing interests

The authors declare no competing interests.

## Additional information

[1]State Key Laboratory of Swine and Poultry Breeding Industry, National Engineering Research Center for Breeding Swine Industry, College of Animal Science, South China Agricultural University, Guangzhou, Guangdong, China. [2]Department of Animal Science, Michigan State University, East Lansing, MI, USA. [3]State Regional Livestock and Poultry Genebank, Guangdong Genebank of Livestock and Poultry, South China Agricultural University, Guangzhou, Guangdong, China. [4]Guangdong Zhongxin Breeding Technology Co., Ltd, Guangzhou, Guangdong, China. [5]College of Animal Science and Technology, Zhongkai University of Agriculture and Engineering, Guangzhou, Guangdong, China. [6]Yunfu Subcenter of Guangdong Laboratory for Lingnan Modern Agriculture, Yunfu, Guangdong, China. [7]Guangdong Provincial Key Laboratory of Agro-animal Genomics and Molecular Breeding, South China Agricultural University, Guangzhou, Guangdong, China. [8]Institute of Animal Science, Guangdong Academy of Agricultural Sciences, Guangdong Key Laboratory of Animal Breeding and Nutrition, Guangzhou, Guangdong, China. [9]These authors contributed equally: Jianping Quan, Ming Yang, Xingwang Wang, Gengyuan Cai ✉e-mail: jieyang2012@hotmail.com; huangw53@msu.edu; wzfemail@163.com

