## [Peer Review file · Nature Communications]

REVIEWER COMMENTS

Reviewer #1 (Remarks to the Author):

This paper “Multi-omic characterization of allele-specific regulatory variation in hybrid pigs” by Jianping Quan et al. is a comprehensive study aimed at understanding gene expression variation in hybrid pigs. The authors analyze genetic variation in DNA sequences and epigenetic modifications to uncover how they contribute to phenotypic variation, particularly focusing on the tissue-specific and developmental stage-specific gene expression in hybrid pigs derived from crosses of Duroc and Lulai breeds. They used a multi-omic approach, including whole genome sequencing, bisulfite sequencing, ATAC-Seq, and RNA-Seq, providing a robust dataset for analyzing allele-specific regulatory variation. The development of a novel read count-based method for assessing allele-specific methylation, chromatin accessibility, and RNA expression is a significant contribution to the field. Creating a high-resolution genome-wide atlas of cis-regulatory variation in pigs across multiple tissues and developmental stages is a pioneering effort in the field. This article is a valuable contribution to the field of genetic research, particularly in understanding allele-specific regulatory variations in pigs. Its strengths lie in its comprehensive approach, innovative methodology, and significant insights. The findings will serve as a vital reference for selecting hybrid breeds and determining hybridization directions in the pig industry.

Generally, allele-specific expression is based on estimates for each SNP, and integrating estimates from different SNPs can introduce significant errors. This study introduces a novel approach to allocate reads to parental alleles, leveraging existing methods based on read counts. To enhance the accuracy of analysis results, the authors also constructed the parental genome for each period, offering significant reference value. The exposition of the manuscript is executed with laudable accuracy and clarity, facilitating comprehension across a multidisciplinary readership.

However, I have several concerns outlined below that need addressing. I believe minor revisions are necessary before considering the manuscript for publication.

Major concerns:

The manuscript would benefit considerably from a more detailed exposition of the figures and tables included. The current descriptions are rather cursory, which detracts from the reader's ability to fully comprehend the presented data. I recommend augmenting these sections with more comprehensive captions or explanatory notes, particularly for the supplementary figures and tables. Additionally, to facilitate smoother cross-referencing, it is imperative to systematically number the figures and tables. For instance, in the supplementary file '476388_0_supp_8480486_s666px.xlsx', the title should explicitly be labeled as 'Table S1', followed

by a descriptive title, such as 'Table S1: The summary of parental genomic sequencing metrics.' Adopting this convention will markedly expedite the review process by allowing for more efficient correlation between the text and the supplementary materials.

Lines 227-244: The expression direction of POE and AGE genes obtained by the authors seems inconsistent across different developmental stages. Further explanation is needed.

Lines 249-259: Among the results (Figure 3c-d), 35 of the 43 imprinted genes identified in previous studies have read counts in this study; 20 genes' expression preference direction is consistent with previous research, while 15 are inconsistent. The authors should elucidate the possible reasons and conduct investigations in ATAC-Seq and BS data.

Lines 318-327: The authors conducted a complete tissue-specific and period-specific analysis of gene expression based on the original genome. However, the chromatin opening level revealed by ATAC-seq data only focused on tissue specificity. Considering the study's experimental design, the authors could also perform more detailed period-specific characterization at the chromatin level. This section needs supplementation.

Minor concerns:

Line 62: Clarify abbreviations like GWAS when first used. Please review and correct all such instances.

Line 91: Change “littermates” to “fullsibs”.

Lines 143, 169, 591: Abbreviations should be defined at their first use and not repeatedly. Please correct.

Lines 154-156, 629, 663: Explain why different alignment software was used for different data categories (BWA for WGS and ATAC-Seq, HISAT2 for RNA-Seq, Bowtie2 for BS data).

Lines 169, 176: Explain the use of TPM for PCA analysis and CPM for edgeR analysis in tissue-specific gene expression. Why not maintain consistency? Please clarify abbreviations.

Lines 184, 194: Justify the definition of "tissue-specific" and "time-specific" expression genes, including a 16-fold higher expression criterion and related references.

Lines 229-232: The authors screened 168 RNA-seq sequenced samples based on the number and proportion of parental origin reads, removing 16 samples whose proportion of parental reads exceeded the 45-55% range. However, in Figure 3, only 6 samples are indicated. Please ensure that the labels for all removed samples are clearly marked in the figure.

Lines 442, 479: Change " $p < 0.05$ " to " $P < 0.05$ " (capitalize and italicize 'P'). Please review the entire text for consistency.

Lines 595-596: Provide a more detailed description of how variation information is integrated into the individual genome.

Lines 699-700, 715-717, 751-752: Provide the model-design formulas directly rather than a literal description.

Reviewer #1 (Remarks on code availability):

I reviewed the code uploaded by the author. The README file is comprehensive and detailed. I installed and executed part of the code, and found it to be accurate and reproducible.

Reviewer #2 (Remarks to the Author):

Remarks to the Author:

Quan and colleagues performed a comprehensive multi-omic characterization of allele-specific regulatory variation in hybrid pigs, using reciprocal crosses. They generated a large dataset of whole genome sequencing, RNA-Seq, ATAC-Seq, and WGBS from four tissues and four developmental stages, and developed a novel read count-based method to assess allele-specific methylation, chromatin accessibility, and expression. They identified genes showing allele-specific expression, parent-of-origin effects, and allele genotype effects, and integrated methylation, chromatin accessibility, and expression data to explain the regulatory mechanisms underlying these effects. The paper is clear and well written and the conclusions rely on a large amount of appropriate

analyses. This study valuable insights into the regulatory landscape and molecular mechanism of gene expression and should reach a wide audience. Nonetheless, I still have few concerns that must be addressed for the work to be published in a journal with the impact of Nature Communications.

Major comments:

1. The authors constructed individualized genomes and transcriptome annotations to overcome the potential reference bias. I personally appreciate such effort. Theoretically, the non-reference genotypes could lead to mapping bias, which have impact on expression quantification and allele specific reads assignment. Nonetheless, the state-of-the-art aligner like HISAT uses graph-based approach to minimize mis-alignment and bias. The authors should provide the comparison of mapping statistics between their approach and common alignment pipelines.
2. The POE genes identified in this study showed a tendency to cluster on chromosomes. The author hypothesize that those genes may be controlled by same cis regulatory element. Previous study by Pan et al (<https://www.nature.com/articles/s41467-021-26153-7>) have detailly annotated the epigenetic landscape in pig. It is necessary and meaningful to check whether there is functional element in close proximity to the POE gene cluster.
3. In most of the figures showing gene expression level, like Fig2c, 2d and Fig7a, read coverages are most enriched at one end of the transcripts. Is it a result of the short isoforms that are easier to meet the criteria for ASE? If so, how did the author deal with potential confounding factors such as transcript length and alternative splicing?
4. One surprising result in this study is that, among the 43 known imprinted genes, only 11 genes were found to be POE genes. Later, based on genome wide chromatin accessibility and CpG methylation signals, the authors hypothesized that the epigenic level can contribute to gene expression variation. The authors should closely examine the allele specific RNA abundance of those known imprinted genes and show whether they support the hypothesis.

Minor points / typos:

1. Line 33 “epigentic” -> “epigenetic”.
2. Line 35 “homogeneous” -> “homogenous”.
3. Line 41 “including 16 whole genome sequenced genomes” -> “including 16 whole genome sequenced individuals”.
4. Line 73 “is proven” -> “has proven
5. Line 87 “Few have ...” -> “Few studies have ...”
6. Line 121 “DNA extracted from ear tissue for all 16 parents” -> “DNA extracted from ear tissue of all 16 parents”

7. Line 139 "the fact that the reference genome was a Duroc pig." -> "the fact that the reference genome was from a Duroc pig."
8. Line 171 "component" -> "component"
9. Line 179 "is" -> "are"
10. Line 182 "in the model" -> "to the model"
11. Line 192 "Tabls S3" -> "Table S3"
12. Line 201 "dynamics" -> "dynamics"
13. Line 302 "Peak heights were quantified within each sample ..." -> "Peak heights were quantified for each sample ..."
14. Line 329 "overlape" -> "overlap"
15. Line 436 "speicifc" -> "specific"
16. Line 478 "and an five of them was" -> "and five of them were"
17. Line 492 "layes" -> "layers"

Reviewer #3 (Remarks to the Author):

The authors described the results of an interesting study that investigated allele-specific expression in crossbred pigs obtained from two different breeds (Duroc and Lulai). The authors investigated different tissues sampled at different developmental stages. A few other studies already reported similar information but using different breeds and not considering multi-omics levels of information.

Line 104: Lulai can be considered a synthetic breed derived by crossing Chinese pigs with Large White pigs.

Lines 163-165: It is not clear what was the impact on removing one sample from one F1 individual who did not cluster with other samples from the same individual and another 9 samples from three F1 individuals (all samples from these individuals) who did not cluster with their littermates (Figure S2). - The error rate in misidentification of the samples might be quite high reducing the reliability of the study.

Line 183: Tissue specific genes might be considered tissue prevalent genes - as the expression was not exclusive for a specific tissue

Line 233: it is not clear how parent-of-origin effect can be better describe the difference between POE and AGE

Figure 6f - the trend lines are difficult to be justified looking at the reported patterns

There are several typos that should be corrected by reading twice the text.

The first section of the results (Experimental design) could be simplified as it is also repeated in Methods.

Reviewer #1:

Q0: This paper “Multi-omic characterization of allele-specific regulatory variation in hybrid pigs” by Jianping Quan et al. is a comprehensive study aimed at understanding gene expression variation in hybrid pigs. The authors analyze genetic variation in DNA sequences and epigenetic modifications to uncover how they contribute to phenotypic variation, particularly focusing on the tissue-specific and developmental stage-specific gene expression in hybrid pigs derived from crosses of Duroc and Lulai breeds. They used a multi-omic approach, including whole genome sequencing, bisulfite sequencing, ATAC-Seq, and RNA-Seq, providing a robust dataset for analyzing allele-specific regulatory variation. The development of a novel read count-based method for assessing allele-specific methylation, chromatin accessibility, and RNA expression is a significant contribution to the field. Creating a high-resolution genome-wide atlas of cis-regulatory variation in pigs across multiple tissues and developmental stages is a pioneering effort in the field. This article is a valuable contribution to the field of genetic research, particularly in understanding allele-specific regulatory variations in pigs. Its strengths lie in its comprehensive approach, innovative methodology, and significant insights. The findings will serve as a vital reference for selecting hybrid breeds and determining hybridization directions in the pig industry.

Generally, allele-specific expression is based on estimates for each SNP, and integrating estimates from different SNPs can introduce significant errors. This study introduces a novel approach to allocate reads to parental alleles, leveraging existing methods based on read counts. To enhance the accuracy of analysis results, the authors also constructed the parental genome for each period, offering significant reference value. The exposition of the manuscript is executed with laudable accuracy and clarity, facilitating comprehension across a multidisciplinary readership.

However, I have several concerns outlined below that need addressing. I believe minor revisions are necessary before considering the manuscript for publication.

Response: We thank the reviewer for a thorough and positive summary and constructive feedback.

Major concerns:

Q1: The manuscript would benefit considerably from a more detailed exposition of the figures and tables included. The current descriptions are rather cursory, which detracts from the reader's ability to fully comprehend the presented data. I recommend augmenting these sections with more comprehensive captions or explanatory notes, particularly for the supplementary figures and tables. Additionally, to facilitate smoother cross-referencing, it is imperative to systematically number the figures and tables. For instance, in the supplementary file '476388_0_supp_8480486_s666px.xlsx', the title should explicitly be labeled as 'Table S1', followed by a descriptive title, such as 'Table S1: The summary of parental genomic sequencing metrics.' Adopting this convention will markedly expedite the review process by allowing for more efficient correlation between the text and the supplementary materials.

Response: We apologize for the omission. Descriptive labels and titles for supplemental tables and captions for supplemental figures are now added.

Q2: Lines 227-244: The expression direction of POE and AGE genes obtained by the authors seems inconsistent across different developmental stages. Further explanation is needed.

Response: We thank the reviewer for their attention to this important aspect of the POE and AGE effects. A small number of POE and AGE genes did exhibit changes in direction of effects across different developmental stages. This is not uncommon. For example, allele specific expression is clearly context dependent at a global scale in mice (Pierre et al., 2022). Furthermore, while we sampled from the same tissues across developmental stages, composition of cell types differed, which may influence POE and AGE effects. We have added additional discussion to the manuscript (Line 497-502).

Reference:

39. Pierre C L S, Macias-Velasco J F, Wayhart J P, et al. Genetic, epigenetic, and environmental mechanisms govern allele-specific gene expression. *Genome research*, 2022, 32(6): 1042-1057.

Line 497-502: "It's notable that a small subset of POE and AGE genes showed a change in direction of their effects across developmental stages. This could be attributed to alteration in the regulation of expression for both POE and AGE genes during development and when context changes, which is not uncommon³⁹. Moreover, tissues at different developmental stages may have different compositions of cell types, potentially leading to variation in the direction of POE and AGE in bulk tissues."

Q3: Lines 249-259: Among the results (Figure 3c-d), 35 of the 43 imprinted genes identified in previous studies have read counts in this study; 20 genes' expression preference direction is consistent with previous research, while 15 are inconsistent. The authors should elucidate the possible reasons and conduct investigations in ATAC-Seq and BS data.

Response: We appreciate this question from the reviewer. We interpret it as one to show that the biallelic expression in RNA is indeed supported by biallelic chromatin accessibility and DNA methylation. This is indeed what we observed for the vast majority of biallelically expressed genes. However, the absence of parent-of-origin effect in chromatin accessibility and DNA methylation near the genes do not exclude other mechanisms that may lead to parent-of-origin effect on RNA expression. That being said, we would also like to point out reasons that are beyond what our data can definitively show and could contribute to the inconsistency between reported imprinting status in databases and our study. First, imprinted genes have tissue and/or developmental stage specific imprinting status. It is not uncommon that two studies in the same species find different imprinted genes simply due to differences in tissues and developmental stages. Second, imprinted gene identification relies on heterozygous genotypes. There may not be alternate alleles in the two breeds that allowed us to identify imprinted genes. Third, many earlier studies may falsely identify imprinted genes based on SNP allele counting, which is known to be biased towards reference

alleles (also see below in response to reviewer 2's Q1). We have largely eliminated this bias by using a read-based counting strategy.

Q4: Lines 318-327: The authors conducted a complete tissue-specific and period-specific analysis of gene expression based on the original genome. However, the chromatin opening level revealed by ATAC-seq data only focused on tissue specificity. Considering the study's experimental design, the authors could also perform more detailed period-specific characterization at the chromatin level. This section needs supplementation.

Response: We thank the reviewer for this suggestion. We have added a comparison of the chromatin accessibility features of stage-specifically expressed genes with their expression levels (Line 331-334), which included additional characterization and a new figure (Figure S8, also see below).

Minor concerns:

Q5: Line 62: Clarify abbreviations like GWAS when first used. Please review and correct all such instances.

Response: Corrected

Q6: Line 91: Change “littermates” to “fullsibs”.

Response: Corrected.

Q7: Lines 143, 169, 591: Abbreviations should be defined at their first use and not repeatedly. Please correct.

Response: Corrected.

Q8: Lines 154-156, 629, 663: Explain why different alignment software was used for different data categories (BWA for WGS and ATAC-Seq, HISAT2 for RNA-Seq, Bowtie2 for BS data).

Response: Our selection of alignment software was guided by the unique characteristics of each data type and the performance and compatibility of the alignment tools. For WGS and ATAC-Seq data, which did not required gapped alignment (to map spliced RNA), we used BWA, which is the industry standard for DNA sequence alignment. For RNA-Seq data, HISAT2 was chosen due to its ability to map RNA to the genome. For bisulfite sequencing, we used Bowtie2 due to the integration of Bowtie2 within the Bismark software, which is a commonly used tool for analyzing methylation data.

Q9: Lines 169, 176: Explain the use of TPM for PCA analysis and CPM for edgeR analysis in tissue-specific gene expression. Why not maintain consistency? Please clarify abbreviations.

Response: For PCA, our goal was to obtain normalized expression levels quickly and accurately. TPM served this purpose well because 1) it was directly output by HISAT2; 2) it was normalized for transcript length and library size; 3) it was easy to combine transcript expression into gene expression.

For edgeR, read counts were needed for the negative binomial model employed by edgeR, hence CPM was used.

Both abbreviations are defined now at first occurrence in the manuscript.

Q10: Lines 184, 194: Justify the definition of “tissue-specific” and “time-specific” expression genes, including a 16-fold higher expression criterion and related references.

Response: We appreciate the opportunity to clarify. The definition of tissue and stage specific genes followed existing literature. We in fact set a more stringent threshold for tissue-specific and stage-specific genes than most other studies. For example, the Human Protein Atlas (HPA) (Uhlén et al. 2015) project defined context (tissue or stage) specific expression as at least 8 fold difference from all other conditions (ours was 16 folds).

Reference:

Uhlen, M. et al. Proteomics. Tissue-based map of the human proteome. Science 347, 1260419, doi:10.1126/science.1260419 (2015).

Q11: Lines 229-232: The authors screened 168 RNA-seq sequenced samples based on the number and proportion of parental origin reads, removing 16 samples whose proportion of parental reads exceeded the 45-55% range. However, in Figure 3, only 6 samples are indicated. Please ensure that the labels for all removed samples are clearly marked in the figure.

Response: We thank the reviewer for this question, which had led us to identify inadvertent omission of a few filtered samples. Labels have now been corrected for Figure S5.

Q12: Lines 442, 479: Change “p<0.05” to “P < 0.05” (capitalize and italicize 'P'). Please review the entire text for consistency.

Response: Corrected.

Q13: Lines 595-596: Provide a more detailed description of how variation information is integrated into the individual genome.

Response: We appreciate the opportunity to clarify. We have added a more detailed description on building individualized genomes (Line 609-612).

Q14: Lines 699-700, 715-717, 751-752: Provide the model-design formulas directly rather than a literal description.

Response: All models are now added to the manuscript.

Q15: I reviewed the code uploaded by the author. The README file is comprehensive and detailed. I installed and executed part of the code, and found it to be accurate and reproducible.

Response: We appreciate the positive feedback on reproducibility.

Reviewer #2:

Q0: Quan and colleagues performed a comprehensive multi-omic characterization of allele-specific regulatory variation in hybrid pigs, using reciprocal crosses. They generated a large dataset of whole genome sequencing, RNA-Seq, ATAC-Seq, and WGBS from four tissues and four developmental stages, and developed a novel read count-based method to assess allele-specific methylation, chromatin accessibility, and expression. They identified genes showing allele-specific expression, parent-of-origin effects, and allele genotype effects, and integrated methylation, chromatin accessibility, and expression

data to explain the regulatory mechanisms underlying these effects. The paper is clear and well written and the conclusions rely on a large amount of appropriate analyses. This study valuable insights into the regulatory landscape and molecular mechanism of gene expression and should reach a wide audience. Nonetheless, I still have few concerns that must be addressed for the work to be published in a journal with the impact of Nature Communications.

Response: We appreciate the thorough and generally positive summary. We address the concerns below in a point-by-point manner.

Major comments:

Q1: The authors constructed individualized genomes and transcriptome annotations to overcome the potential reference bias. I personally appreciate such effort. Theoretically, the non-reference genotypes could lead to mapping bias, which have impact on expression quantification and allele specific reads assignment. Nonetheless, the state-of-the-art aligner like HISAT uses graph-based approach to minimize mis-alignment and bias. The authors should provide the comparison of mapping statistics between their approach and common alignment pipelines.

Response: We thank the reviewer for this important comment. We have conducted additional evaluation for our read-based methods as well as commonly used SNP based methods. We tested two approaches in a randomly selected representative sample. First, we used our newly developed read-based counting to partition reads into those that mapped to the Duroc sow or the Lulai boar by mapping reads to either genome separately. Second, we mapped reads to the reference genome (an unrelated Duroc animal) and counted reads supporting either the reference or non-reference allele using GATK's ASEReadCounter, a commonly used approach for estimating ASE. The results are summarized below.

- Mapping to individualized genomes improved mapping rate. 7.28% of reads were either unmapped or mapped to non-unique positions when mapping to a single reference genome. However, 6.66% of reads remained unmapped or mapped to non-unique positions when mapping to two individualized genomes.
- We have shown that the percentages of reads mapped to the paternal and maternal genomes were generally similar without appreciable bias (Figure S5).
- On a per-SNP basis, it's not possible to count reads originated from the paternal or maternal genome. Nevertheless, we compared the proportion of reads mapped to the maternal allele for each SNP and the proportion of reads mapped to the maternal genome for the gene where the SNP resides (Figure below). Two strong patterns emerged:
 1. There are multiple SNPs per gene, and there is substantial variation among the SNPs. This presents a challenge to integrate multiple SNPs within a gene to make gene level inference.
 2. There is a strong bias towards unequal allele representation by SNP-based allele counts than by gene-based read counts. Other than agreement along the diagonal, the slope of the points is substantially smaller than 1, suggesting that SNP based analysis tends to significantly

overestimate allele bias.

These results strongly indicate that our read-based approach is sufficient to overcome the bias introduced by SNP-based analysis.

Q2: The POE genes identified in this study showed a tendency to cluster on chromosomes. The author hypothesize that those genes may be controlled by same cis regulatory element. Previous study by Pan et al (<https://www.nature.com/articles/s41467-021-26153-7>) have detailly annotated the epigenetic landscape in pig. It is necessary and meaningful to check whether there is functional element in close proximity to the POE gene cluster.

Response: We thank the revier for this suggestion. We have now compared the positions of the POE genes with the functional regulatory elements (enhancers, promoters, and inhibitors) identified by Pan *et al.* Our analysis revealed that 24, 68 and 79 POE genes overlapped with enhancer regions in brain, liver and muscle, respectively. Fiver, 16 and 29 POE genes overlapped with promoter regions in brain, liver and muscle, respectively. And 6, 2 and 35 POE genes overlapped with repressor regions in brain, liver and muscle, respectively.

Additionally, we found that one inhibitor and two enhancer regions overlapped with two POE genes.

This information has also been added as a supplemental Table 7 to the paper.

	Overlapped POE gene number			Elements regulate multiple POE genes		
	Enhancer	Promoter	Repressor	Enhancer	Promoter	Repressor
Brain	24	5	6	0	0	0
Liver	68	16	2	2	0	0
Muscle	79	29	35	0	0	1

Q3: In most of the figures showing gene expression level, like Fig2c, 2d and Fig7a, read coverages are most enriched at one end of the transcripts. Is it a result of the short isoforms that are easier to meet the

criteria for ASE? If so, how did the author deal with potential confounding factors such as transcript length and alternative splicing?

Response: We thank the reviewer for the opportunity to clarify. Our ASE read counting were performed on constitutive non-overlapping exons only so presumably the results are not influenced by alternative splicing. The enrichment of read coverage at the end of transcripts in ASE analysis (Figure 7a) is likely due to the fact that informative variants happened to be located near the 3' end. This is not unexpected because 3'UTRs harbor more DNA variants than coding sequences. The enrichment or read coverage towards one end versus the other in the analysis of tissue and stage specificity (Figure 2) does not appear to be 3' specific. For example, the gene *MYOD1* seems to attract reads on the 5' end while *IGFN1* on the 3' end. There can be many reasons including but are not limited to library preparation bias. Given that the gene *MYOD1* has only one exon, it's unlikely that these end enrichment is due to alternative splicing.

Q4: One surprising result in this study is that, among the 43 known imprinted genes, only 11 genes were found to be POE genes. Later, based on genome wide chromatin accessibility and CpG methylation signals, the authors hypothesized that the epigenetic level can contribute to gene expression variation. The authors should closely examine the allele specific RNA abundance of those known imprinted genes and show whether they support the hypothesis.

Response: As we have responded previously to Q3 of reviewer 1, imprinted gene status is context dependent and thus it's not surprising that only 11 imprinted genes were validated. We thank the reviewer for the suggestion to look closely in our data for concordance between RNA-Seq and chromatin accessibility and methylation variation, which is in fact beyond of the scope of this paper as a global characterization. Nevertheless, we randomly selected 4 genes, namely *SGCE*, *MEST*, *DLK1*, and *NDN*, and found a general negative correlation between the methylation levels of potential regulatory regions of these genes and gene expression. In addition, we visualized the concordance between methylation, chromatin accessibility and RNA expression, all of which showed POE effects for the gene *SGCE* (Figure S13). This has also been included in a new supplemental figure 13.

Minor points / typos:

Q5: Line 33 “epigentic” > “epigenetic”.

Response: Corrected.

Q6: Line 35 “homogeneous” > “homogenous”.

Response: Corrected.

Q7: Line 41 “including 16 whole genome sequenced genomes” > “including 16 whole genome sequenced individuals”.

Response: Corrected.

Q8: Line 73 “is proven” > “has proven

Response: Corrected.

Q9: Line 87 “Few have ...” > “Few studies have ...”

Response: Corrected.

Q10: Line 121 “DNA extracted from ear tissue for all 16 parents” > “DNA extracted from ear tissue of all 16 parents”

Response: Corrected.

Q11: Line 139 “the fact that the reference genome was a Duroc pig.” > “the fact that the reference genome was from a Duroc pig.”

Response: Corrected.

Q12: Line 171 “component” > “component”

Response: Corrected.

Q13: Line 179 “is” > “are”

Response: Corrected.

Q14: Line 182 “in the model” > “to the model”

Response: Corrected.

Q15: Line 192 “Tabls S3” > “Table S3”

Response: Corrected.

Q16: Line 201 “dynamics” > “dynamics”

Response: Corrected.

Q17: Line 302 “Peak heights were quantified within each sample ...” > “Peak heights were quantified for

each sample ...”

Response: Corrected.

Q18: Line 329 “overlape” > “overlap”

Response: Corrected.

Q19: Line 436 “speicifc” > “specific”

Response: Corrected.

Q20: Line 478 “and an five of them was” > “and five of them were”

Response: Corrected.

Q21: Line 492 “layes” > “layers”

Response: Corrected.

Reviewer #3:

Q0: The authors described the results of an interesting study that investigated allele-specific expression in crossbred pigs obtained from two different breeds (Duroc and Lulai). The authors investigated different tissues sampled at different developmental stages. A few other studies already reported similar information but using different breeds and not considering multi-omics levels of information.

Response: We thank the reviewer for the summary and constructive feedbacks, which we find particularly helpful.

Q1: Line 104: Lulai can be considered a synthetic breed derived by crossing Chinese pigs with Large White pigs.

Response: This information is now added to the manuscript: “Duroc is a major commercial breed known for its superior growth performance, while Lulai is a synthetic breed derived from crosses between Laiwu pigs and Large White pigs, which possess desirable meat quality traits, including high intramuscular fat content”.

Q2: Lines 163-165: It is not clear what was the impact on removing one sample from one F1 individual who did not cluster with other samples from the same individual and another 9 samples from three F1 individuals (all samples from these individuals) who did not cluster with their littermates (Figure S2). - The error rate in misidentification of the samples might be quite high reducing the reliability of the study.

Response: Misidentification and sample contamination is not uncommon in genomic studies. We applied stringent criteria to remove 10 samples (6%) that did not pass our filter. We completely agree with the

reviewer that misidentification and/or contamination can lead to downstream problems in data analysis. However, we consider the 6% filter rate reasonably low. We believe removing these problematic samples in fact increase realibility of the study.

Q3: Line 183: Tissue specific genes might be considered tissue prevalent genes - as the expression was not exclusive for a specific tissue

Response: We appreciate the opportunity to clarify. Tissue specificity is generally not used to imply tissue exclusive. Our definition of tissue and stage specific genes followed existing literature. We in fact set a more stringent threshold for tissue-specific and stage-specific genes than most other studies. For example, the Human Protein Atlas (HPA) (Uhlén et al. 2015) project defined context (tissue or stage) specific expression as at least 8 fold difference from all other conditions (ours was 16 folds).

Reference:

Uhlen, M. et al. Proteomics. Tissue-based map of the human proteome. Science 347, 1260419, doi:10.1126/science.1260419 (2015).

Q4: Line 233: it is not clear how parent-of-origin effect can be better describe the difference between POE and AGE

Response: Figure 1e was specifically used to illustrate the difference between POE and AGE. The parent-of-origin effect (POE) was defined as the effect by the maternal or paternal origin of the allele regardless of allele type, and the allele genotype effect (AGE) was defined as the effect by the breed origin of the allele. To clarify further, POE captures the distinction between alleles inherited from the mother versus the father, while AGE represents the differences between alleles derived from different breeds. Our reciprocal cross design allowed us to distinguish the two.

Q5: Figure 6f - the trend lines are difficult to be justified looking at the reported patterns

Response: The original Figure 6f had points overlapping each other so density of points was unclear. In the revised manuscript, we have enhanced the clarity of Figure 6f by representing the density of point distributions using color gradients, highlighting the relationship between gene expression levels and the methylation level at regulatory regions. The modified figure is as follows:

Q6: There are several typos that should be corrected by reading twice the text.

Response: We have carefully read through the entire text and made corrections to typos.

Q7: The first section of the results (Experimental design) could be simplified as it is also repeated in Methods.

Response: We appreciate this suggestion. However, we feel a strong need to retain the description of experimentl design in the results section to provide important context for subsequent description of results.

REVIEWERS' COMMENTS

Reviewer #1 (Remarks to the Author):

I have carefully reviewed the revisions and the responses provided to the comments on the manuscript titled "Multi-omic characterization of allele-specific regulatory variation in hybrid pigs" by Jianping Quan et al. I am pleased to see that all the concerns raised during the initial review have been comprehensively addressed. The enhancements made to the figures and tables, along with the addition of detailed descriptions and systematic numbering, have significantly improved the clarity and comprehensibility of the manuscript. Given the substantial improvements and the rigorous efforts to meet the high standards of the journal, I am satisfied with the current state of the manuscript.

Reviewer #2 (Remarks to the Author):

Quan and colleagues have addressed my major concerns. I believe that the revised version of the manuscript is considerably improved as a result of these efforts.

Last but not the least, I still have one concern that the author should consider further elaborating. The authors clearly stated the advantages of the ASE methods used in this study. But the limitation and applicability are lacking. For example, the reported approach can identify cis regulated genes but provide no insights into the regulatory mechanisms with respect to DNA variants. The authors should at least discuss these limitations.

Reviewer #3 (Remarks to the Author):

The manuscript has been adjusted following what was suggested. It would be useful to include a reference to support what was stated at lines 106 and 107.

Reviewer #1 (Remarks to the Author):

Q1: I have carefully reviewed the revisions and the responses provided to the comments on the manuscript titled "Multi-omic characterization of allele-specific regulatory variation in hybrid pigs" by Jianping Quan et al. I am pleased to see that all the concerns raised during the initial review have been comprehensively addressed. The enhancements made to the figures and tables, along with the addition of detailed descriptions and systematic numbering, have significantly improved the clarity and comprehensibility of the manuscript. Given the substantial improvements and the rigorous efforts to meet the high standards of the journal, I am satisfied with the current state of the manuscript.

Response: We thank the reviewer for the positive feedback.

Reviewer #2 (Remarks to the Author):

Q1: Quan and colleagues have addressed my major concerns. I believe that the revised version of the manuscript is considerably improved as a result of these efforts. Last but not the least, I still have one concern that the author should consider further elaborating. The authors clearly stated the advantages of the ASE methods used in this study. But the limitation and applicability are lacking. For example, the reported approach can identify cis regulated genes but provide no insights into the regulatory mechanisms with respect to DNA variants. The authors should at least discuss these limitations.

Response: We appreciate the reviewer for the positive summary and for this suggestion. We have added additional discussion about limitation and applicability of the allele specific expression approach used in the revised manuscript (Line 499-503).

Line 499-503: While the design in this study can discover genes under cis regulation and potentially reveal the regulatory cascade, it offers no information on the regulatory mechanisms with respect to DNA variants. Further high resolution mapping using population scale datasets, high throughput reporter assays, and gene editing are required to causally delineate the cis regulatory mechanisms.

Reviewer #3 (Remarks to the Author):

Q1: The manuscript has been adjusted following what was suggested. It would be useful to include a reference to support what was stated at lines 106 and 107.

Response: We thank the reviewer for this suggestion. We have added the references in the revised manuscript (Line 100-101).

Line 100-101: Duroc is a major commercial breed known for its superior growth performance, while Lulai is a synthetic breed derived from crosses between Laiwu pigs and Large White

pigs²⁰, which possess desirable meat quality traits, including high intramuscular fat content²¹.

Reference:

20 Cao, R. et al. Genomic signatures reveal breeding effects of Lulai pigs. *Genes* 13, 1969 (2022).

21 Yan, M. et al. Investigation on muscle fiber types and meat quality and estimation of their heritability and correlation coefficients with each other in four pig populations. *Animal Science Journal* 95, e13915 (2024).